# The RhoGAP activity of CYK-4/MgcRacGAP functions non-canonically by promoting RhoA activation during cytokinesis

**Donglei Zhang, Michael Glotzer\***

Department of Molecular Genetics and Cell Biology, University of Chicago, Chicago, United States

**Abstract** Cytokinesis requires activation of the GTPase RhoA. ECT-2, the exchange factor responsible for RhoA activation, is regulated to ensure spatiotemporal control of contractile ring assembly. Centralspindlin, composed of the Rho family GTPase-activating protein (RhoGAP) MgcRacGAP/CYK-4 and the kinesin MKLP1/ZEN-4, is known to activate ECT-2, but the underlying mechanism is not understood. We report that ECT-2-mediated RhoA activation depends on the ability of CYK-4 to localize to the plasma membrane, bind RhoA, and promote GTP hydrolysis by RhoA. Defects resulting from loss of CYK-4 RhoGAP activity can be rescued by activating mutations in ECT-2 or depletion of RGA-3/4, which functions as a conventional RhoGAP for RhoA. Consistent with CYK-4 RhoGAP activity contributing to GEF activation, the catalytic domains of CYK-4 and ECT-2 directly interact. Thus, counterintuitively, CYK-4 RhoGAP activity promotes RhoA activation. We propose that the most active form of the cytokinetic RhoGEF involves complex formation between ECT-2, centralspindlin and RhoA.

## Introduction

### RhoA activation during cytokinesis

Cell division involves the myosin-mediated contraction of an actin-based contractile ring. In metazoans, contractile ring assembly involves activation of the GTPase RhoA. RhoA directly activates formin-mediated actin polymerization and indirectly promotes myosin activation. As the contractile ring must assemble at the correct position and at the correct time, between the segregating chromosomes in anaphase, RhoA activation is subject to multiple regulatory mechanisms (see *Green et al., 2012* for review).

The primary direct activator of RhoA during cytokinesis is the RhoGEF ECT-2. ECT-2 contains N-terminal BRCT domains and a C-terminal RhoGEF domain (*Kim et al., 2005*; *Zou et al., 2014*). Presumably as a consequence of being autoinhibited, ECT-2 function depends on activators. One of the important activators of ECT-2 during cytokinesis is the centralspindlin complex, which is a heterotetramer containing a dimeric kinesin, ZEN-4 (aka MKLP1, Pavarotti) and dimeric CYK-4 (aka MgcRacGAP, Tum/RacGAP50C) (*Mishima et al., 2002*). Centralspindlin organizes the spindle midzone and directly recruits numerous regulators of cytokinesis, including ECT-2, to this location (*Burkard et al., 2009*; *Wolfe et al., 2009*). ECT-2 and centralspindlin are conserved among—and restricted to—metazoans (*Frédéric et al., 2013*) (unpublished results). Though these proteins are conserved, their names are distinct in each organism. For simplicity, *Caenorhabditis elegans* names will be used throughout this manuscript with the exception that we will refer to RHO-1 with the more common name RhoA.

**\*For correspondence:**
mglotzer@uchicago.edu

**Competing interests:** The authors declare that no competing interests exist.

**Reviewing editor**: Mohan Balasubramanian, University of Warwick, United Kingdom

**eLife digest** Cell division is a process in which a cell splits to form two daughter cells. In most cases, the cell first duplicates its genetic material and then the two copies are pulled to opposite ends of the cell. A ring of protein filaments—called the contractile ring—then assembles to form a band around the cell at the site of the division. This ring contracts and the force generated separates the cells in a step known as cytokinesis.

A protein belonging to the Rho family, called RhoA, is essential for cytokinesis because it controls the formation of the contractile ring. Rho proteins are switched on by the activities of other proteins called guanine nucleotide exchange factors. Another group of proteins known as 'GTPase activating proteins' (or GAPs for short) generally act to promote the ability of Rho proteins to turn themselves off.

In animals and other multicellular organisms, a GAP called CYK-4 largely concentrates on the spindle midzone, but some of the protein also moves to part of the cell membrane near the future site of cell division. It binds to a guanine nucleotide exchange factor called ECT-2 to switch RhoA on, which in turn promotes the formation of the contractile ring. However, it is not clear why a protein that activates RhoA is also able to trigger its inactivation.

In this study, Zhang and Glotzer studied cell division in a roundworm called *Caenorhabditis elegans*. The experiments show that cells that lacked the GAP activity of CYK-4 were unable to complete cytokinesis because RhoA was not fully switched on. This requirement could be bypassed in cells with mutant forms of ECT-2 that were overactive. Therefore, an activity that was thought to inactivate RhoA actually promotes its activation. Further experiments show that the section (or 'domain') of CYK-4 that has GAP activity interacts directly with the guanine nucleotide exchange domain of ECT2. Zhang and Glotzer suggest that this interaction stimulates ECT2 and thereby promotes the activation of RhoA.

Further experiments will reveal how CYK-4 stimulates ECT-2. In addition, it will be important to determine whether other proteins with GAP domains also work in this unconventional way.

Recruitment of ECT-2 to the spindle midzone involves regulated binding between ECT-2 and CYK-4. The BRCT domains of ECT-2 bind to CYK-4 phosphorylated by PLK-1 (*Burkard et al., 2009*; *Wolfe et al., 2009*). CYK-4 phosphorylation occurs in a cell cycle and microtubule-regulated manner. Furthermore, CDK-1 phosphorylation of ECT-2 inhibits the ECT-2-CYK-4 interaction during metaphase and inactivates a membrane binding motif within ECT-2 (*Yüce et al., 2005*; *Su et al., 2011*). The phosphorylation-dependent interaction between ECT-2 and centralspindlin is required for RhoA activation during cytokinesis in human cells (*Burkard et al., 2007*; *Wolfe et al., 2009*).

Centralspindlin also localizes in trace, but biologically relevant, amounts on the cell membrane. Centralspindlin accumulation to the midzone and the membrane are independently regulated by its oligomerization, which is inhibited by a 14-3-3 protein and promoted by the chromosome passenger complex (*Douglas et al., 2010*; *Basant et al., 2015*).

## The role of the GAP activity of CYK-4 GAP has been controversial

The CYK-4 subunit of centralspindlin contains an evolutionarily conserved Rho family GTPase-activating protein (RhoGAP) domain (*Jantsch-Plunger et al., 2000*). The function of this domain has been examined in a number of different contexts. In vitro, the GAP domain of CYK-4 efficiently activates the GTPase activity of the Rho-family GTPases, CED-10/Rac1 and CDC-42. CYK-4 also has GAP activity towards RhoA, but it is far less active towards RhoA as compared to Rac1 and CDC-42 (*Touré et al., 1998*; *Jantsch-Plunger et al., 2000*; *Bastos et al., 2012*). Despite extensive effort, there is no consensus for the biological role of this GAP activity (see *White and Glotzer, 2012* for review). In some cell types, the GAP activity appears dispensable (*Goldstein et al., 2005*; *Yamada et al., 2006*), in others it appears to be important to negatively regulate Rac1 (*D'Avino et al., 2004*; *Canman et al., 2008*; *Bastos et al., 2012*), whereas in yet others it appears to promote RhoA activation (*D'Avino et al., 2004*; *Zavortink et al., 2005*; *Loria et al., 2012*). System-specific differences may underlie some of these diverse results. However, these studies differ in the mutations used to assess the function of the GAP domain, which is likely to affect the results. In addition, some of the controversy may be due to misinterpretation of indirect effects.

The function of the RhoGAP domain has been examined in *C. elegans* embryos in some detail. These studies have focused largely on a temperature-sensitive, separation-of-function substitution mutation, E448K, that lies in the RhoGAP domain of CYK-4, *cyk-4(or749ts)* (*Canman et al., 2008*). The mutant protein can complex with the centralspindlin kinesin, ZEN-4, and bundle microtubules in the central spindle. However, cytokinesis does not proceed to completion in these embryos. Interestingly, depletion of CED-10/Rac1 or the actin nucleator subunit ARP-2 enables these embryos to complete cytokinesis (*Canman et al., 2008*). These genetic interactions have been interpreted to indicate that the GAP domain of CYK-4 is important to keep CED-10/Rac1 inactive and prevent the accumulation of branched actin in the equatorial region (*Canman et al., 2008*). However, subsequent analysis demonstrated that *cyk-4(or749ts)*; *ced-10(−)* embryos are phenotypically abnormal (*Loria et al., 2012*). In particular, *cyk-4*; *ced-10* mutant embryos have reduced accumulation of RhoA effectors as compared to *ced-10* mutants alone (*Loria et al., 2012*). These results suggest that this mutation in CYK-4 affects more than the RhoGAP activity of CYK-4 or that CED-10/Rac1 is not the relevant target of the GAP domain, or both.

## Parallel mechanisms activate RhoA in the early *C. elegans* embryo

Analysis of centralspindlin function in *C. elegans* embryos is impeded by the existence of a second, parallel pathway that promotes RhoA activation in the early embryo. Upon fertilization, embryos exhibit RhoA-dependent contractility that promotes embryo polarization and culminates in the formation of a transient furrow known as the pseudocleavage furrow. This wave of contractility requires a poorly conserved protein known as NOP-1 and is largely independent of centralspindlin (*Tse et al., 2012*). Cytokinetic contractility, on the other hand, involves both NOP-1 and centralspindlin. However, *nop-1* is nonessential; loss of function mutants are viable and fertile (*Rose et al., 1995*). Cytokinesis proceeds to completion in the absence of NOP-1, although furrow initiation is slightly delayed and RhoA accumulates to lower levels at the cleavage furrow (*Tse et al., 2012*). Due to its role in RhoA activation, polarization in NOP-1-deficient embryos is also delayed. Mutational inactivation of NOP-1 permits direct analysis of centralspindlin-dependent furrow formation. Notably, when NOP-1 is inactivated, *cyk-4(or749ts)* embryos are completely defective in RhoA activation (*Tse et al., 2012*).

## Membrane-bound centralspindlin promotes RhoA activation

The CYK-4 GAP domain is adjacent to a C1 domain that mediates membrane localization of centralspindlin (*Lekomtsev et al., 2012*). Given that the *cyk-4(or749ts)* substitution renders the protein thermosensitive, it is possible that these phenotypes are not the sole consequence of loss of GAP activity; this mutation could affect other functions of the GAP domain, the adjacent C1 domain may also be affected. To clarify these issues, we performed a targeted structure-function analysis of the C1 and GAP domains of CYK-4. We demonstrate that the *cyk-4(or749ts)* allele indeed affects its ability to associate with the membrane and show that this activity contributes to RhoA activation. We further show that the active site of the GAP domain contributes to the accumulation of downstream effectors of RhoA and RhoA-dependent contractility. Furthermore, we find that the catalytic domains of CYK-4 and ECT-2 directly interact in vitro. Finally, we show that hypomorphic defects in CYK-4-mediated RhoA dependent contractility can be suppressed by either loss of the RhoGAP activity provided by RGA-3/4 or by either of two activating mutations in ECT-2. These activating mutations in *ect-2* rescue *cyk-4(or749ts)* and GAP-deficient CYK-4. Our results indicate that CYK-4 GAP activity is involved in ECT-2-mediated RhoA activation.

## Results

### The *cyk-4(or749ts)* allele, E448K, exhibits defects in membrane association

We sought to conduct a structure-function analysis of CYK-4 to determine the individual contributions of the GAP and C1 domains of CYK-4. We established a rescue assay based on single copy integrants of GFP-tagged CYK-4 transgenes driven by the *cyk-4* promoter inserted at a defined locus in the *C. elegans* genome using Mos1-mediated integration (*Figure 1—figure supplement 1*) (*Frøkjaer-Jensen et al., 2012*). The transgenes were expressed at consistent levels (*Figure 1—figure supplement 2*).

The transgenes were rendered resistant to an RNAi construct that could effectively deplete endogenous CYK-4 by targeting the 3′ UTR and portions of the coding sequence (*Figure 1—figure supplement 2*). By combining the appropriate mutant transgene with RNAi to specifically deplete endogenous CYK-4, we obtained embryos that express CYK-4$^{MUT}$. In this and all subsequent experiments, when a given variant is assayed, endogenous CYK-4 is depleted by RNAi; these will be referred to as *cyk-4$^{mut}$* embryos. We first sought to validate the rescue assay, by assaying *cyk-4$^{E448K}$* embryos and found that they closely phenocopy *cyk-4(or749ts)* embryos in which endogenous CYK-4 has the E448K substitution (*Figure 1—figure supplement 3*). This phenocopy indicates functional depletion of endogenous CYK-4. Additionally, a wild-type transgene was fully functional as it could complement a large deletion in CYK-4 to viability and fertility (*Figure 6—figure supplement 1*).

Consistent with the C1 domain promoting membrane association, CYK-4$^{ΔC1}$ does not accumulate on ingressing cleavage furrows (*Figure 1A,B*). As *cyk-4(or749ts)* is temperature sensitive, we considered the possibility that the thermosensitivity also destabilizes the C1 domain that lies adjacent to the CYK-4 GAP domain (*Figure 1—figure supplement 1*). Indeed, at the restrictive temperature, CYK-4$^{E448K}$ exhibits a similar defect in localization as CYK-4$^{ΔC1}$ (*Figure 1A,B*). CYK-4 also associates with the membrane in the germline and indirect evidence suggests that this localization is compromised in embryos expressing CYK-4$^{E448K}$ (*Zhou et al., 2013*). We generated strains in which both the endogenous *cyk-4* and the GFP-tagged transgene contained the E448K mutation. At the permissive temperature, CYK-4$^{E448K}$ membrane recruitment was readily detected in the germline, however this localization was lost as animals were shifted to the restrictive temperature (*Figure 1—figure supplement 4*). Similarly, the C1 domain is required for membrane localization in the gonad (*Figure 1—figure supplement 4*). Thus, CYK-4$^{E448K}$ impairs membrane localization, and it is likely to impair the function of both its GAP and C1 domains.

We compared the progression of cytokinesis in embryos expressing CYK-4$^{WT}$, CYK-4$^{ΔC1}$, and CYK-4$^{E448K}$. CYK-4$^{ΔC1}$ embryos exhibit a cytokinetic defect that is largely similar to that of *cyk-4$^{E448K}$* embryos. In particular, furrow ingression is slow and incomplete (*Figure 1C,F*). However, subtle differences were detected; the onset of significant furrow ingression is delayed relative to controls in *cyk-4$^{E448K}$* embryos but not in *cyk-4$^{ΔC1}$* embryos. This suggests that the CYK-4$^{E448K}$ phenotype could reflect a compound defect, rather than a sole defect in the ability to associate with the membrane.

Cleavage furrow ingression in *C. elegans* depends on the combined action of centralspindlin and a non-essential protein, NOP-1 (*Tse et al., 2012*). To determine whether the C1 domain is essential for centralspindlin-dependent furrow ingression, we expressed CYK-4$^{WT}$, CYK-4$^{ΔC1}$, and CYK-4$^{E448K}$ in embryos that lack the NOP-1-dependent pathway for furrow ingression. As expected, CYK-4$^{WT}$ supports full furrow ingression in this sensitized background. In stark contrast, neither CYK-4$^{ΔC1}$ nor CYK-4$^{E448K}$ support detectable furrow ingression in the absence of NOP-1 activity (*Figure 1D,G*).

Previous studies demonstrated that loss of function mutations in *ced-10/rac-1* or depletion of the protein partially suppress the cytokinesis defect in *cyk-4(or749ts)* embryos (*Figure 1E,H*). However, as described above, it is important to examine the extent of suppression in the absence of the parallel, NOP-1-dependent, pathway. We therefore examined whether *ced-10/rac-1* loss of function mutations could suppress the cytokinesis defect in *cyk-4$^{ΔC1}$*. Although cleavage furrows in *cyk-4$^{ΔC1}$*; *ced-10(n1993)* embryos ingress somewhat more deeply than *cyk-4$^{ΔC1}$* embryos, they do not complete cytokinesis (*Figure 1E,H*). The simplest interpretation of these results is that although CYK-4$^{E448K}$ diminishes membrane association of CYK-4, it may retain some function at the restrictive temperature such that it facilitates the abscission step in *cyk-4(or749ts)*; *ced-10(n1993)* embryos, as CYK-4-mediated membrane association is essential for completion of cytokinesis in cultured human cells (*Lekomtsev et al., 2012*). Importantly, when NOP-1 activity is compromised, inactivation of CED-10/Rac1 does not suppress the defect in furrow ingression caused by CYK-4$^{E448K}$ (*Figure 1J,I*), suggesting that CYK-4 does not act directly on CED-10.

## Mutants in the active site of the CYK-4 GAP domain are cytokinesis-defective

To begin to determine the role of the catalytic activity of CYK-4 during cytokinesis, we first studied the consequence of mutating the highly conserved catalytic arginine that stabilizes the transition state during GTP hydrolysis. Substitution of the catalytic arginine with alanine strongly attenuates GAP activity against Rac in a variety of CYK-4 orthologs and is widely used to inactivate Rho family GAPs

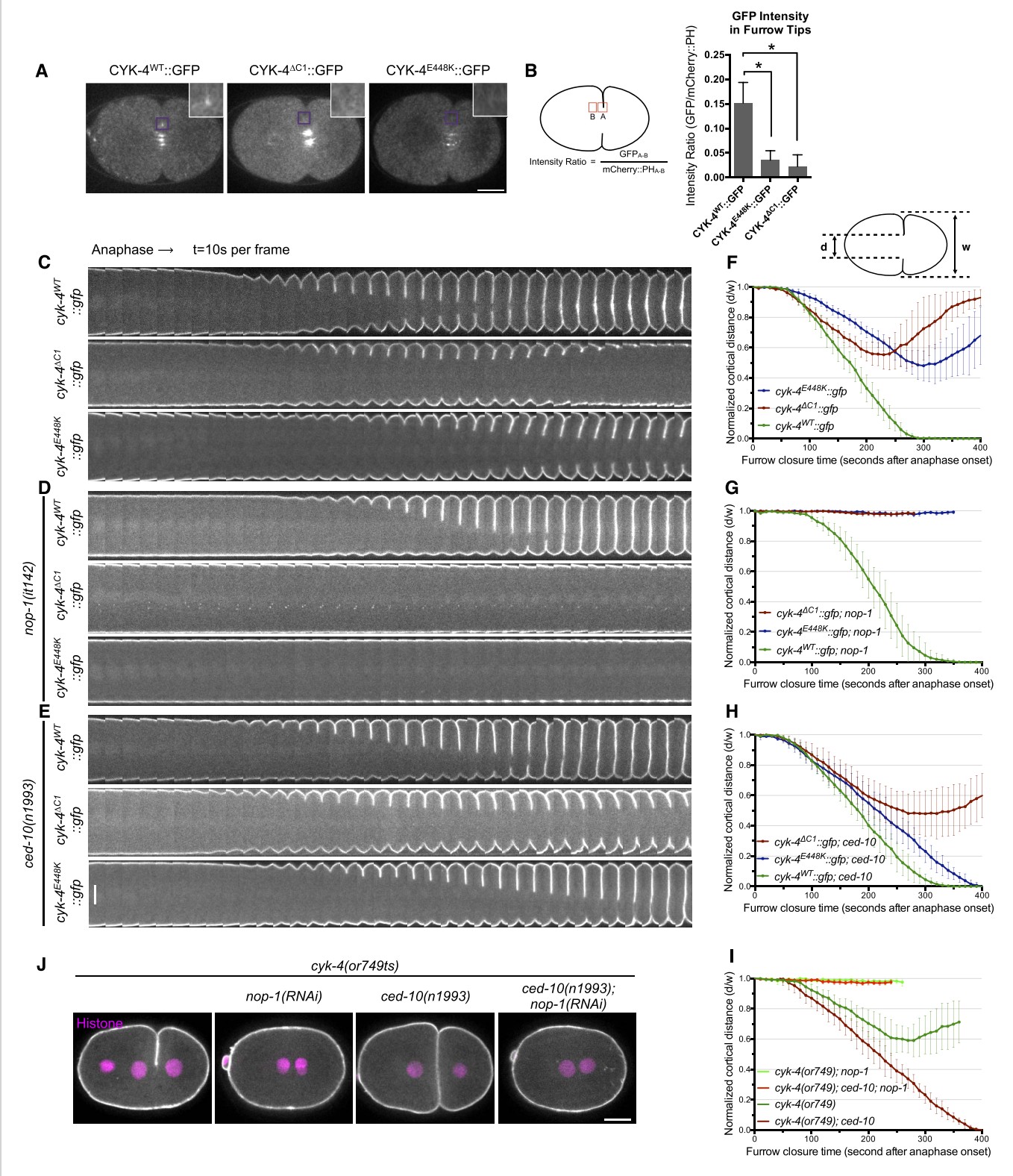

**Figure 1**. CYK-4-dependent membrane binding promotes furrow ingression. (**A**) CYK-4 accumulates on the plasma membrane. Membrane accumulation is observed on ingressing cleavage furrows (boxed regions). CYK-4 membrane accumulation requires the C1 domain and is compromised by the E448K substitution in the *cyk-4(or749ts)* allele. (**B**) Membrane accumulation of CYK-4 variants. The accumulation was quantified as a ratio of the accumulation of CYK-4::GFP/mCherry::PH at the furrow tip as depicted in the schematic; the mean intensity ±s.e.m. are plotted. (N = 8–12 embryos *p < 0.05, one way

*Figure 1. continued on next page*

*Figure 1. Continued*

ANOVA followed by Tukey multiple comparison). (**C**) Deletion of the C1 domain and the E448K substitution impair cleavage furrow ingression. Kymographs generated from time-lapse recordings of the equatorial region of embryos of the indicated genotypes expressing the membrane marker mCherry::PH. Kymographs begin at anaphase onset. (**D**) Deletion of the C1 domain and the E448K substitution abrogate centralspindlin-dependent furrow ingression. The progression of cytokinesis was assessed in embryos expressing CYK-4 variants in combination with a loss of function mutation in *nop-1*. (**E**) Mutation of CED-10/Rac1 slightly increases furrow ingression in CYK-4$^{\Delta C1}$ embryos and allows complete, albeit delayed, furrow ingression in CYK-4$^{E448K}$ embryos. The progression of cytokinesis was assessed in embryos expressing CYK-4 variants in combination with a loss of function mutation in *ced-10*. (**F–I**) Quantification of furrow ingression rates in embryos of the indicated genotypes. Representative examples are shown in the accompanying kymographs. (N = 8–12 embryos; error bars, 95% confidence intervals). (**J**) The ability of *ced-10/Rac1* to rescue cytokinesis in CYK-4$^{E448K}$ embryos requires the NOP-1 pathway for furrow ingression. Images shown reflect the maximal extent of furrow ingression in embryos of the indicated genotypes expressing the membrane marker mCherry::PH. Unless otherwise specified, all scale bars in all figures are 10 µm.

The following figure supplements are available for figure 1:

**Figure supplement 1**. CYK-4 structure, mutations, and methods of transgene integration.

**Figure supplement 2**. Transgene expression levels.

**Figure supplement 3**. The CYK-4$^{E448K}$::GFP transgene combined with *cyk-4(RNAi)* closely phenocopies *cyk-4(or749ts)*.

**Figure supplement 4**. Assay for localization of CYK-4 variants to the membrane of the gonad.

(*Rittinger et al., 1997*; *Yamada et al., 2006*; *Miller and Bement, 2009*; *Bastos et al., 2012*; *Zanin et al., 2013*). As expected, CYK-4$^{R459A}$ GAP domain retains the ability to bind to RhoA•GTP, demonstrating that the protein is well folded in vitro (*Figure 2—figure supplement 1*). While CYK-4 GAP exhibits GAP activity towards both RhoA and CED-10/Rac1, CYK-4$^{R459A}$ GAP lacks detectable GAP activity towards either GTPase (*Figure 2—figure supplement 2*). In order to determine if catalytic activity is required for viability, we used a strain heterozygous for a deletion mutant of CYK-4, *cyk-4(ok1034)*. One quarter of the embryos from these heterozygous hermaphrodites contain maternally provided CYK-4 and lack zygotic CYK-4. These zygotic null embryos fail to hatch and arrest with a variety of terminal phenotypes (*Figure 2A*, left). Many, but not all, embryos contain muscle tissue and have undergone partial morphogenesis. The embryos also contain enlarged cells, likely due to defects in cytokinesis (*Sugimoto et al., 2001*). We introduced the GAP-defective transgene into this strain. Remarkably, *cyk-4(ok1034)*; *cyk-4$^{R459A}$* animals hatch and develop to adulthood. However, these animals are sterile (*Figure 2A*, middle). Thus, the GAP activity of CYK-4 is not essential for post embryonic development but it has an important role in gonad development, likely due to a requirement for post-embryonic cell proliferation in the germline. As a consequence, it is not possible to use classical genetic tools to obtain embryos in which CYK-4$^{R459A}$ is the sole form of CYK-4.

To study the role of the CYK-4 GAP activity during embryogenesis, we used the aforementioned assay (*Figure 1—figure supplement 1*). Embryos, expressing only CYK-4$^{R459A}$, are largely normal during the initial stages of the first cell cycle. They undergo pseudocleavage, mitotic spindle assembly, chromosome segregation, central spindle assembly during anaphase, and CYK-4$^{R459A}$ becomes highly enriched on the spindle midzone (*Figure 2B*). Cleavage furrow initiation occurs and the furrow ingresses at near wild-type rates to near completion. However, cytokinesis does not complete and the furrow ultimately regresses; this phenotype was fully penetrant (*Figure 2C*); these embryos also fail to complete cytokinesis following meiosis II (data not shown). These results suggest that a late step in cytokinesis is most sensitive to loss of CYK-4 GAP activity. This phenotype is distinct from that of *cyk-4(or749ts)* embryos–cleavage furrows in *cyk-4$^{R459A}$* embryos ingress more rapidly and more deeply than *cyk-4$^{E448K}$* embryos (*Figure 1C*). We assessed the ability of CYK-4$^{R459A}$ to associate with membrane during furrow ingression (*Figure 2B,D*). CYK-4$^{R459A}$ hyper accumulates on the membrane as compared to WT CYK-4; this localization suggests that CYK-4R459A is well folded in vivo. Therefore, the cytokinetic defect in this strain is unlikely to be an indirect consequence of a failure of CYK-4 to localize to the membrane.

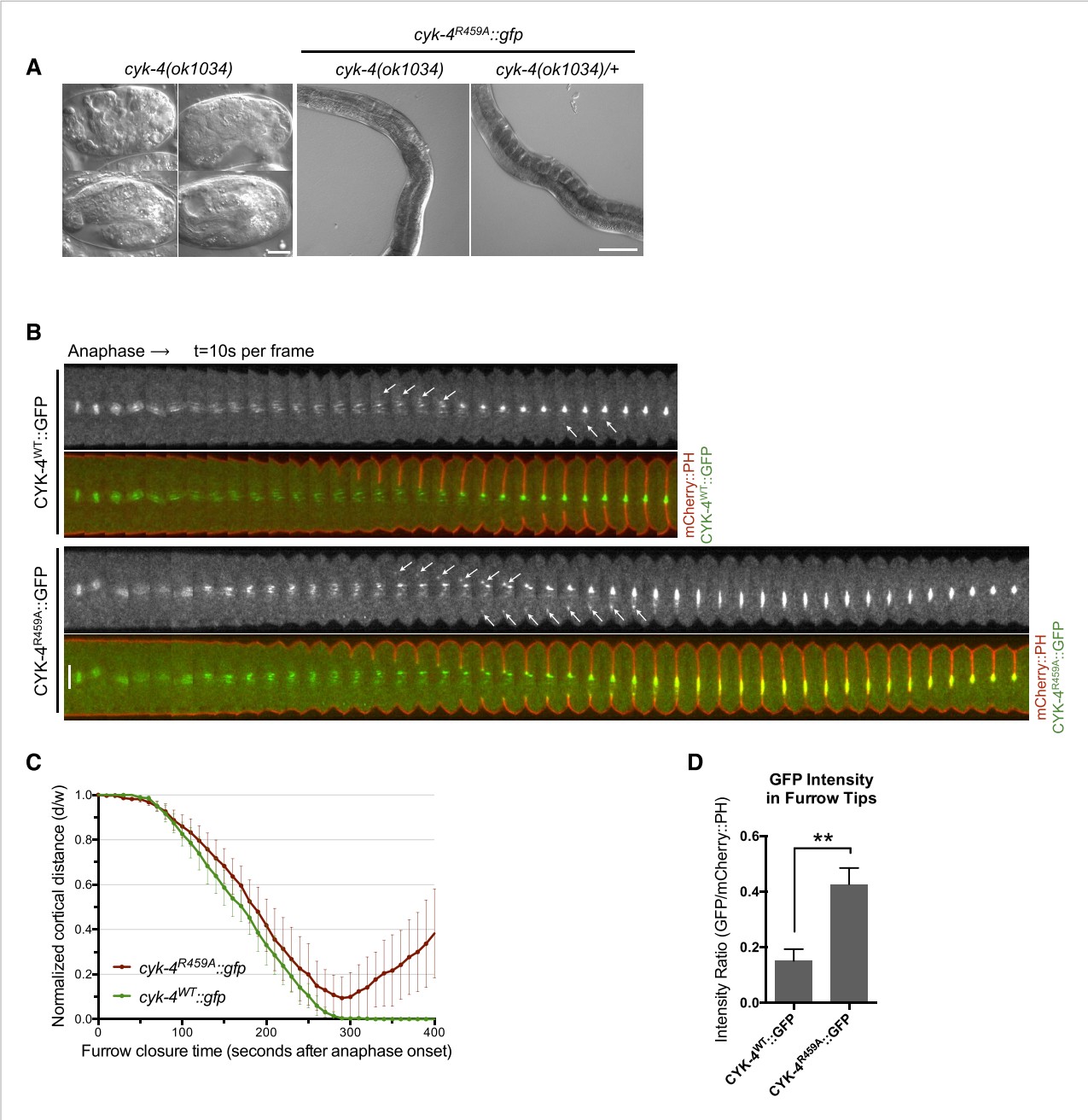

**Figure 2**. CYK-4 GAP activity is required for cytokinesis and viability. (**A**) *cyk-4(ok1034)* null embryos arrest at variable stages during embryogenesis (left). The GAP-deficient CYK-4$^{R459A}$ transgene rescues the embryonic lethality, but animals arrest as sterile adults (middle). GAP-deficient CYK-4$^{R459A}$ is recessive, no phenotypes is seen in the presence of *cyk-4(+)* (right). Right scale bar 50 μm. (**B**) GAP-deficient CYK-4$^{R459A}$ embryos fail to complete cytokinesis. Kymographs are generated as in **Figure 1D**, with the exception that the signal from the CYK-4 transgenes is also shown (green), overlaid on mCherry::PH (red). CYK-4$^{R459A}$ accumulates more strongly at furrow tips as compared to CYK-4$^{WT}$ (arrows). (**C**) The kinetics of furrow ingression in CYK-4$^{WT}$ and CYK-4$^{R459A}$ embryos. Results are quantified as described in **Figure 1G**. (**D**) Membrane accumulation of CYK-4$^{WT}$ and CYK-4$^{R459A}$. Results are quantified as described in **Figure 1B** (**p < 0.01, by t-test).

The following figure supplements are available for figure 2:

**Figure supplement 1**. Binding of CYK-4 variants to RhoA.

**Figure supplement 2**. GAP activity of CYK-4 and variants.

## CYK-4 GAP domain mutations that prevent RhoA binding are highly defective in RhoA activation

To extend these results, confirm that CYK-4 must interact with Rho family GTPases during cytokinesis, and eliminate the possibility that the phenotype of CYK-4$^{R459A}$ is due to enhanced binding of CYK-4 to active RhoA, we engineered mutations in CYK-4 that reduce its binding to RhoA and other GTPases (*Rittinger et al., 1997*; *Sekimata et al., 1999*). Two conserved, surface exposed, basic residues in the RhoA interface (K495, R499) (*Figure 3—figure supplement 1*) were charge reversed to glutamic acid, generating CYK-4$^{EE}$, and characterized in the transgenic rescue assay. Embryos expressing only CYK-4$^{EE}$, like those expressing CYK-4$^{R459A}$ and CYK-4$^{\Delta C1}$, exhibit fully penetrant embryonic lethality (*Figure 3—figure supplement 1*). CYK-4$^{EE}$ exhibits reduced binding to RhoA in vitro (*Figure 2—figure supplement 1*), and it does not exhibit membrane hyperaccumulation in vivo (*Figure 3A,B*). Interestingly, *cyk-4$^{EE}$* embryos exhibit a stronger furrow ingression defect than *cyk-4$^{R459A}$* embryos, as furrow ingression is slower and less complete (*Figure 3C,D*). Importantly, NOP-1 depletion from *cyk-4$^{EE}$* embryos largely eliminates furrow ingression (*Figure 3C,D*). Thus, Rho GTPase binding by CYK-4 is essential for centralspindlin-mediated cytokinetic ingression.

We next sought to determine the Rho family GTPase to which CYK-4 must bind to fulfill its function in vivo. If furrow formation is dependent on CYK-4 binding to either CED-10/Rac1 or CDC-42 to generate a positive regulatory complex, then inactivation of these GTPases would be predicted to cause a phenotype at least as severe as a mutation that weakens the GTPase binding site of CYK-4. However, mutation of CED-10/Rac1, or depletion of CDC-42, does not affect the rate of cleavage furrow ingression (*Jantsch-Plunger et al., 2000*; *Loria et al., 2012*), even when combined with mutations in NOP-1 (*Figure 3E,F*). Indeed, cytokinesis occurs efficiently and proceeds to completion in embryos in which NOP-1, CED-10/Rac1, and CDC-42 are simultaneously inactivated (*Figure 3E,F*). We infer, therefore, that RhoA is the relevant GTPase that CYK-4 binds to promote cleavage furrow formation. Due to its direct role in furrow ingression, it is not possible to test RhoA in the same manner.

## CYK-4 GAP domain mutations are defective in centralspindlin-dependent furrowing

We next sought to determine how the GAP activity of CYK-4 promotes cytokinesis. Previous studies proposed at least three models for the phenotype seen in *cyk-4$^{R459A}$* embryos. First, the GAP domain could function as canonical GAP that acts on RhoA, causing CYK-4 GAP-deficient embryos fail to complete cytokinesis because of a requirement for RhoA inactivation at late cytokinesis. Second, CED-10/Rac1 could be an important target of CYK-4 GAP activity, causing CYK-4 GAP deficient embryos to accumulate ectopic Rac1 activity that interferes with cytokinesis. Third, although it is counterintuitive, CYK-4 GAP activity could somehow promote RhoA activation, and therefore the CYK-4 GAP deficient embryos may fail cytokinesis due to incomplete RhoA activation. We sought to distinguish between these alternatives.

The first and third models make opposite predictions for the outcome of experiments in which RhoA levels are perturbed (*Figure 4A*). If the failure to complete cytokinesis in *cyk-4$^{R459A}$* embryos is due to hyperactivation of RhoA, as would be predicted from the canonical model for the function of a RhoA GAP, then a reduction in active RhoA levels might ameliorate the defect. Conversely, if the GAP active site promotes RhoA activation, then the reduction of RhoA activity would be predicted to exacerbate the phenotype of *cyk-4$^{R459A}$*. To distinguish between these models, we reduced RhoA levels by mutationally inactivating NOP-1. As expected, all control embryos (*nop-1(it142)*; *cyk-4$^{WT}$*) complete cytokinesis (*Figure 4Bi*). Surprisingly, *nop-1(it142)*; *cyk-4$^{R459A}$* mutant embryos exhibit extremely weak furrow ingression; furrows in these embryos ingressed less than ~10% of egg width (*Figure 4Biii,C*). This result supports models in which CYK-4 GAP activity is involved in RhoA activation.

## *ced-10(−)* only partially bypasses loss of CYK-4 GAP activity

Previous studies have implicated CED-10/Rac1 as a target of CYK-4 GAP activity, although these studies utilized the *cyk-4(or749ts)* mutation that impairs membrane localization of CYK-4 (*Figure 1A, B*, *Figure 1—figure supplement 4A*). Therefore, we addressed whether loss of function mutations in *ced-10* affect cytokinesis in *cyk-4$^{R459A}$* embryos. Interestingly, we found that all *ced-10*; *cyk-4$^{R459A}$*

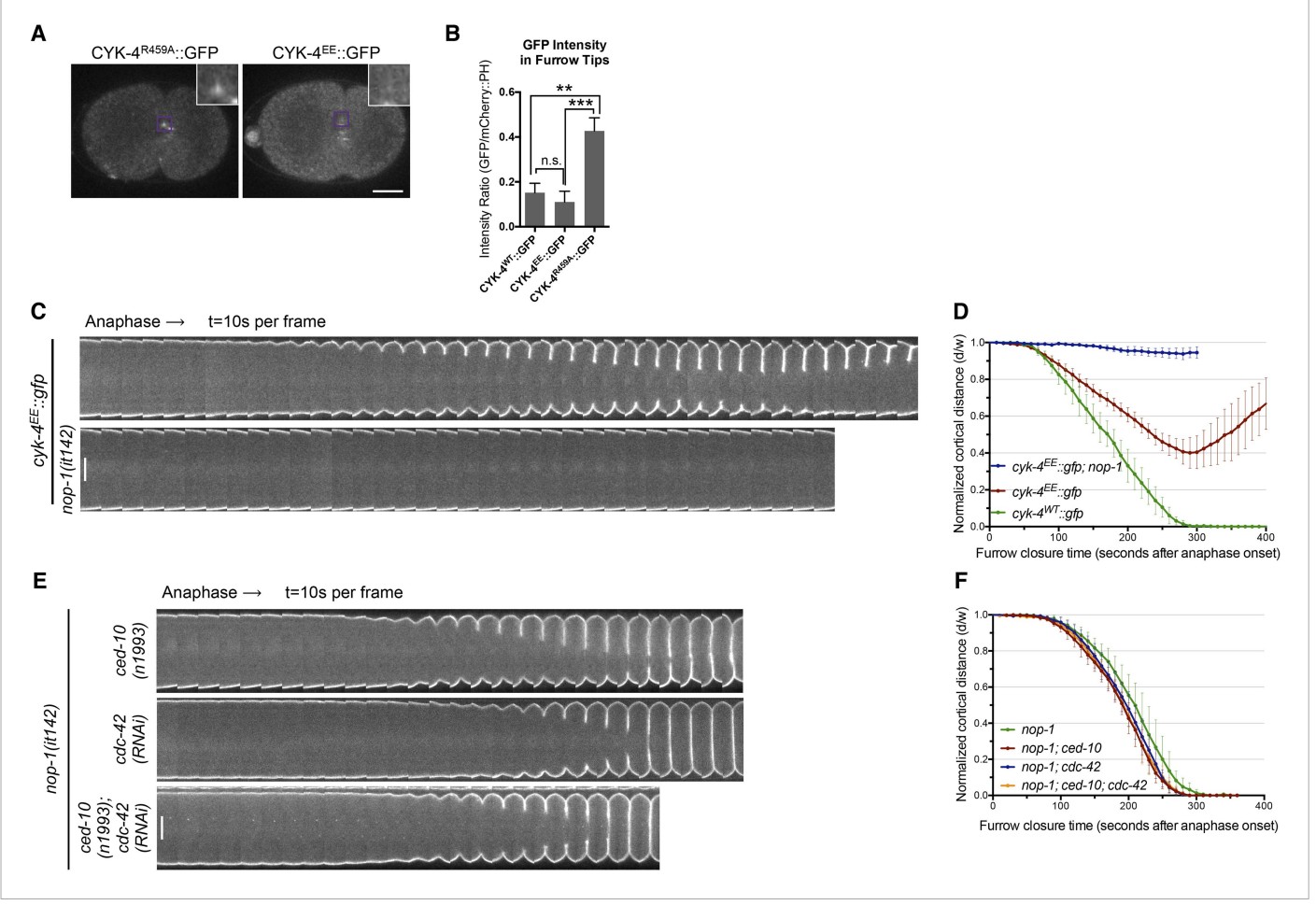

**Figure 3**. RhoA binding by CYK-4 is required for cytokinesis. (**A**) CYK-4$^{R459A}$, but not CYK-4$^{EE}$, hyperaccumulates on the plasma membrane of the ingressing cleavage furrows (boxed regions). (**B**) Membrane accumulation of CYK-4::GFP variants was quantified as described in *Figure 1B*. (N = 8–12 embryos; **p < 0.01, by one way ANOVA followed by Tukey multiple comparison). (**C**) The Rho family GTPase binding defective variant of CYK-4, CYK-4$^{EE}$, causes cytokinesis defects. Kymograph analysis of the progression of cytokinesis in CYK-4$^{EE}$ embryos in the presence or absence of NOP-1 function. Kymographs were assembled as described in legend to *Figure 1D*. (**D**) The kinetics of furrow ingression in CYK-4$^{EE}$ embryos. Results are quantified as described in *Figure 1G*. (**E**) Inactivation of CDC-42 and CED-10/Rac1, either alone or in combination, does not cause cytokinesis defects in the sensitized *nop-1* background. Kymographs were assembled as described in legend to *Figure 1D*. Note that depletion of CDC-42 results in symmetric cleavage furrow ingression. (**F**) The kinetics of furrow ingression in embryos deficient in *nop-1* and/or CDC-42 and/or CED-10/Rac1 function. Results are quantified as described in *Figure 1G*.

The following figure supplement is available for figure 3:

**Figure supplement 1**. Location and conservation of mutated residues in the CYK-4 GAP domain.

embryos fully ingress and 75% complete cytokinesis (*Figure 4Biv,C*), suggesting significant, albeit incomplete rescue. Depletion of ARX-2, a component of the Arp2/3 complex, a downsteam effector of Rac GTPases, provides similar rescue as mutation in *ced-10* (*Figure 4—figure supplement 1*). Two other Rac related proteins, RAC-2 and MIG-2, could, in principle, be additional targets of the CYK-4 GAP domain. However, depletion of RAC-2 does not rescue completion of cytokinesis in *cyk-4$^{R459A}$* embryos (*Figure 4—figure supplement 1*). Furthermore, gain of function mutations in *mig-2* (*Zipkin et al., 1997*) do not cause cytokinesis defects, even in sensitized genetic backgrounds (*Figure 4—figure supplement 2*).

Mutations in *ced-10* also slightly increase the extent of furrow ingression in *cyk-4$^{ΔC1}$* embryos (*Figure 1E,H*). To more stringently test whether the GAP activity of CYK-4 is linked with CED-10/Rac1

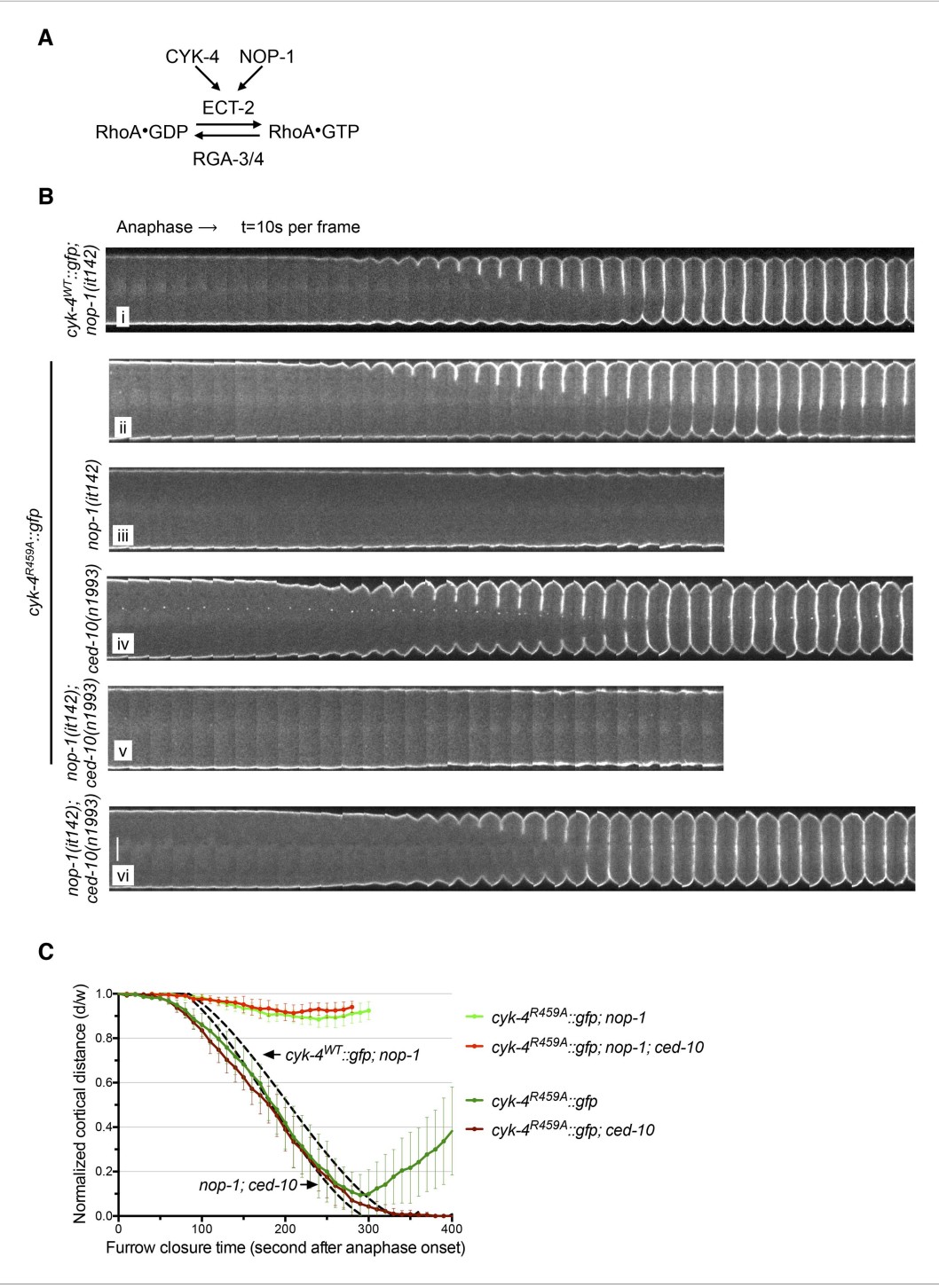

**Figure 4**. The GAP activity of CYK-4 is required for centralspindlin-dependent furrowing independent of CED-10/Rac1. (**A**) Schematic depiction of the known regulators of RhoA. (**B**) GAP defective CYK-4 causes cytokinesis defects that are greatly enhanced by loss of NOP-1 function; this defect is not fully suppressed by mutation of *ced-10/rac1*. Kymograph analysis of the progression of cytokinesis in CYK-4$^{R459A}$ embryos in the presence or absence of CED-10/Rac1 and/or NOP-1 function. The kinetics of furrow ingression in *cyk-4$^{WT}$::gfp; nop-1(it142)* and *nop-1(it142); ced-10(n1993)* embryos are shown for comparison. Kymographs were assembled as described in legend to **Figure 1D**. (**C**) The kinetics of furrow ingression in CYK-4$^{R459A}$ embryos. Results are quantified as described in **Figure 1G**. The kinetics of furrow ingression in *cyk-4$^{WT}$::gfp; nop-1(it142)* and *nop-1(it142); ced-10(n1993)* embryos from **Figures 1F, 3F** are shown as dashed lines for comparison.

*Figure 4. continued on next page*

*Figure 4. Continued*

The following figure supplements are available for figure 4:

**Figure supplement 1**. Depletion of ARX-2, but not RAC-2, suppresses the cytokinesis defect in CYK-4$^{R459A}$ embryos.

**Figure supplement 2**. A gain of function mutation, *mig-2(gm103)*, does not affect cytokinesis in sensitized backgrounds.

**Figure supplement 3**. Depletion of neither ARX-2 nor RAC-2 modulates the cytokinesis defect in *nop-1; ced-10; cyk-4$^{R459A}$* embryos.

inactivation, we assessed the progression of cytokinesis in embryos that lack NOP-1 function. Crucially, *nop-1; ced-10* embryos complete cytokinesis (*Figure 4Bvi*). If CED-10/Rac1 inactivation is the primary function of the CYK-4 GAP domain, then CYK-4 GAP activity would be predicted to be dispensable in *nop-1; ced-10* embryos. However, in stark contrast to this prediction, *nop-1; ced-10; cyk-4$^{R459A}$* embryos fail to form ingressing cleavage furrows altogether (*Figure 4Bv,C*). Significant furrow ingression is not restored by depletion of either RAC-2 or ARX-2 in *nop-1; ced-10; cyk-4$^{R459A}$* embryos, suggesting that the cytokinesis defect is not due to activation of other Rac-family proteins (*Figure 4—figure supplement 3*). These data demonstrate that the catalytic activity of the CYK-4 GAP domain must have a function that is distinct from maintaining CED-10/Rac1 in an inactive state.

## Mutations in the active site of CYK-4 can be suppressed by RGA-3/4 depletion

To further test models in which CYK-4 RhoGAP catalytic activity is important to either promote RhoA activation or to promote RhoA inactivation, we examined the consequence of depletion of the predominant RhoA GAP in the early embryo, RGA-3/4 (*Schmutz et al., 2007*; *Schonegg et al., 2007*). As previously shown, depletion of RGA-3/4 causes cortical hypercontractility in otherwise wild-type embryos, during both pseudocleavage and cytokinesis, and results in embryonic lethality (*Figure 5A*) (*Schmutz et al., 2007*; *Schonegg et al., 2007*). When RGA-3/4 is depleted from *cyk-4$^{R459A}$* embryos, all embryos complete cytokinesis (*Figure 5B*, *Figure 5—figure supplement 1A*), further suggesting that the GAP activity of CYK-4 promotes, rather than counteracts, RhoA activation.

To test this model more stringently, we asked whether cytokinesis also completes when RGA-3/4 is depleted from *cyk-4$^{R459A}$* embryos also lacking NOP-1 (i.e., *nop-1(it142); rga-3/4(RNAi); cyk-4$^{R459A}$* embryos). Remarkably, although furrows in *cyk-4$^{R459A}$; nop-1* embryos barely ingress, when RGA-3/4 is depleted, furrow ingression is completed in 100% of embryos (*Figure 5B,C*, *Figure 5—figure supplement 1B*). This result also rules out the possibility that RGA-3/4 depletion allows completion of cytokinesis because it stabilizes RhoA that was activated in a NOP-1-dependent manner. Depletion of RGA-3/4 did not significantly modify the cytokinetic phenotype of *nop-1; cyk-4(RNAi)* embryos (*Figure 5C*, *Figure 5—figure supplement 1B*), demonstrating CYK-4 dependence to this suppression. Furthermore, complete suppression was specific to *cyk-4$^{R459A}$* embryos, depletion of RGA-3/4 induced deeper but still incomplete ingression in *nop-1; cyk-4$^{EE}$* and *nop-1; cyk-4$^{\Delta C1}$* embryos. These strains formed an allelic series in order of decreasing extents of ingression: *cyk-4$^{R459A}$ > cyk-4$^{EE}$ > cyk-4$^{\Delta C1}$ ∼ cyk-4(RNAi)* (*Figure 5C*, *Figure 5—figure supplement 1B*).

## Mutations in the GAP domain of CYK-4 prevent accumulation of RhoA effectors

RhoA is a dose dependent regulator of cleavage furrow formation (*Loria et al., 2012*) and CYK-4 is involved in RhoA activation by relieving autoinhibition of ECT-2 (*Kim et al., 2005*; *Yüce et al., 2005*). We therefore assayed whether CYK-4 GAP domain mutations affect accumulation of RhoA effectors. Because the RhoA biosensor and the CYK-4 transgenes are both GFP-tagged and integrated at the same position of the genome, we assayed the accumulation of RFP-tagged non-muscle myosin, NMY-2, a key effector of RhoA, as a proxy for RhoA activation. To validate that NMY-2::mRFP is a valid proxy for RhoA activity levels, we compared the accumulation of these two markers to the cleavage furrow during cytokinesis when co-expressed. The recruitment of NMY-2::mRFP and the RhoA biosensor are

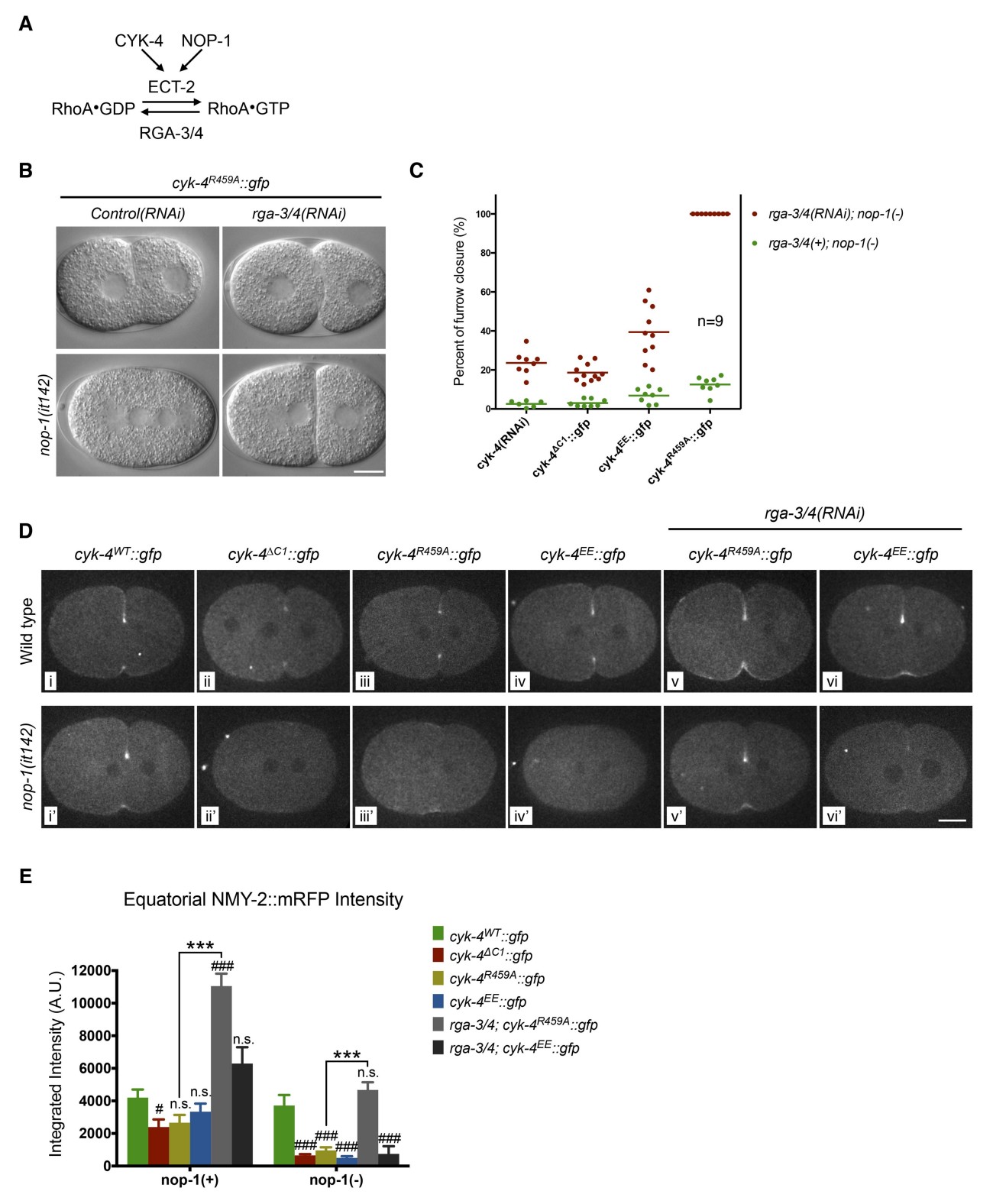

**Figure 5**. Depletion of RGA-3/4 rescues cytokinesis in CYK-4$^{R459A}$ embryos. (**A**) Schematic depiction of the known regulators of RhoA. (**B**) Representative embryos demonstrating the effect of RGA-3/4 depletion on cytokinesis in CYK-4$^{R459A}$ embryos both in the presence and absence of NOP-1 function. (**C**) Depletion of RGA-3/4 rescues cytokinesis specifically in CYK-4$^{R459A}$ embryos. CYK-4$^{\Delta C1}$, CYK-4$^{EE}$, CYK-4$^{R459A}$ where expressed in *nop-1(it142)* embryos and the extent of furrow closure measured either in the presence (green) or the absence (red) of RGA-3/4. (**D**) Accumulation of the RhoA effector NMY-2

*Figure 5. continued on next page*

*Figure 5. Continued*

(tagged with RFP) in embryos of the indicated genotypes. Embryos are shown at ~50% (or maximal) ingression. Note the reduction of cortical myosin accumulation in CYK-4$^{R459A}$, CYK-4$^{EE}$, CYK-4$^{\Delta C1}$ embryos as compared to CYK-4$^{WT}$ (i–iv). The severity of this reduction is enhanced by inactivation of NOP-1 (i'–iv'). Depletion of RGA-3/4 restores myosin accumulation in CYK-4$^{R459A}$ and CYK-4$^{EE}$ embryos (v, vi), even in embryos defective in NOP-1 function (v', vi'). (**E**) Quantification of total NMY-2::mRFP accumulation in the furrow region over the course of cytokinesis in embryos of the indicated genotypes. Error bars, s.e.m.; n.s. (not significant); #p < 0.05; ###p < 0.001 refers to significance relative to wild-type in *nop-1(+)* and *nop-1(it142)*, respectively by one way ANOVA followed by Tukey multiple comparison. ***p < 0.001 for the indicated comparison.

The following figure supplements are available for figure 5:

**Figure supplement 1**. Furrow ingression of RGA-3/4-depleted embryos expressing CYK-4 variants.

**Figure supplement 2**. Comparison between accumulation of NMY-2::RFP and the RhoA biosensor during cytokinesis.

highly correlated in space, time, and intensity (*Figure 5—figure supplement 2A,B*). In addition, the correlation between these markers remains strong when either NOP-1, CYK-4, or RGA-3/4 are depleted, despite the significant changes in the extent of recruitment caused by these perturbations. Thus NMY-2::mRFP provides a reliable proxy for RhoA activation.

We assayed NMY-2::mRFP levels in CYK-4$^{WT}$, CYK-4$^{\Delta C1}$, CYK-4$^{R459A}$, and CYK-4$^{EE}$ embryos during anaphase. Mutations in CYK-4 that reduce the rate and extent of cleavage furrow also reduce NMY-2::mRFP accumulation (*Figure 5D*, top row, *Figure 5E*). The defect in myosin accumulation caused by mutations in the GAP domain of CYK-4 is far more severe and apparent in NOP-1-depleted embryos (*Figure 5D*, bottom row, *Figure 5E*). Conversely, depletion of RGA-3/4 increases myosin accumulation in CYK-4$^{R459A}$ and CYK-4$^{EE}$ embryos, both in the presence and absence of NOP-1. These data support models in which the catalytic activity of the CYK-4 GAP domain contributes to RhoA activation.

## CYK-4 GAP domain mutations can be suppressed by gain of function mutations in *ect-2*

To obtain additional insight into the mechanism by which CYK-4 promotes cytokinesis, we took an unbiased genetic suppression approach. We mutagenized *cyk-4(or749ts)* animals, grew the mutagenized animals at the permissive temperature for two generations to allow potential suppressors to become homozygous and shifted them to 25°C to select for suppressors. We isolated three strong suppressors out of a total of ~105 mutagenized F1 genomes. Suppressor strains were subjected to sequencing of the *cyk-4* locus to identify potential intragenic suppressors. One strain contained a substitution mutation in CYK-4, H485Y, relatively close to the *or749ts* substitution E448K (*Figure 6A*). We also isolated two strong extragenic suppressors, *xs110* and *xs111*, that rescue *cyk-4 (or749ts)* to viability at the restrictive temperature. The suppressed strains complete cytokinesis with high efficiency (>90%) and support high viability (*Figure 6—figure supplement 1*).

Candidate extragenic suppressors were genetically mapped to chromosome II near the *ect-2* locus. Substitution mutations in the *ect-2* locus were identified in both suppressor strains (*Figure 6A*). *ect-2 (xs110)* contained a single nucleotide change in the PH domain, resulting in a G707D substitution (*Figure 6A*, *Figure 6—figure supplement 2A*). In a related RhoGEF for which there is a co-crystal structure with RhoA (PDZRhoGEF), the residue analogous to G707 lies in an α helix in the PH domain that comes into close proximity to the α3 helix of RhoA (*Figure 6—figure supplement 2B*) (*Chen et al., 2010*).

We recorded the progression of cytokinesis in *cyk-4(or749ts); ect-2(xs110)* embryos and found that the embryos not only complete cytokinesis as expected but also the delay in furrow initiation and the slow furrow ingression phenotypes characteristic of *cyk-4(or749ts)* embryos were largely corrected (*Figure 6B,Cii*). Thus, unlike *ced-10(n1993)*, *ect-2(xs110)* suppresses the primary defect of the *cyk-4 (or749ts)* mutation.

To confirm that the *ect-2(xs110)* substitution was causative, we used the CRISPR-associated nuclease Cas9 to re-create this mutation (*Zhang and Glotzer, 2014*). We injected *cyk-4(or749ts)* animals with a plasmid that expresses both Cas9 and a sgRNA designed to create a double strand break near E705 and provided an oligonucleotide repair template containing the G707D substitution.

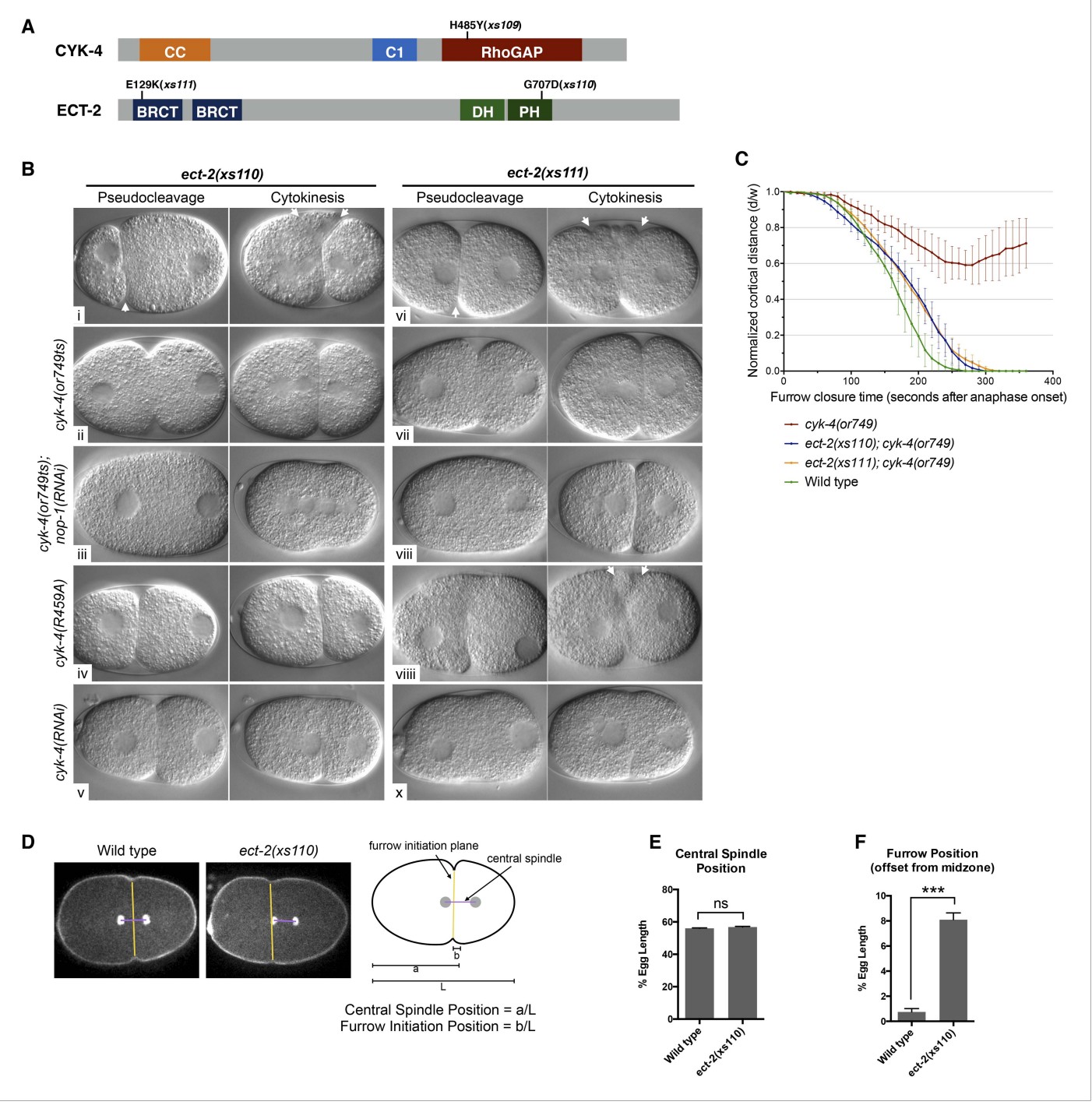

**Figure 6.** Mutations in ECT-2 suppress *cyk-4(or749ts)*. (**A**) Schematic depicting the domain structure of CYK-4 and ECT-2 with the positions of the mutations isolated in the *cyk-4(or749ts)* suppressor screen. (**B**) *ect-2(xs110)* and *ect-2(xs111)* suppresses both *cyk-4(or749ts)* and CYK-4^R459A. Images of embryos of the indicated genotypes are shown at the pronuclear stage and during cytokinesis. Both *ect-2(xs110)* (i) and *ect-2(xs111)* (v) embryos exhibit hypercontractility (arrows) that is suppressed by *cyk-4(or749ts)* (ii and vii). Depletion of NOP-1 from *cyk-4(or749ts); ect-2(xs110)* eliminates contractility during pseudocleavage and greatly reduces contractility during cytokinesis (iii). Depletion of NOP-1 from *cyk-4(or749ts); ect-2(xs111)* eliminates contractility during pseudocleavage but cytokinesis is still observed (viii). *ect-2(xs110)* and *ect-2(xs111)* both allow cytokinetic completion in CYK-4^R459A (iv, ix). Depletion of CYK-4 prevents completion of cytokinesis in *ect-2(xs110)* and *ect-2(xs111)* (v, x). Phenotypes shown were seen in (i) 18/18 embryos; (ii) 11/12; (iii) 7/16, 5/16 showed less contractility; (iv) 6/6; (v) 7/7; (vi) 14/14; (vii) 18/18; (viii)13/16; (ix) 6/6; (x) 6/6. (**C**) *ect-2(xs110)* and *ect-2(xs111)* suppress *cyk-4 (or749ts)*. The kinetics of furrow ingression in *cyk-4(or749ts)* embryos and in the suppressed strains. Results are quantified as described in *Figure 1G*. (**D**) *ect-2(xs110)* causes defects in cleavage plane positioning. The position of furrow initiation and the spindle midzone are indicated in yellow and purple,

*Figure 6. Continued*

respectively (see schematic). (**E**) Quantification of the mean position of the central spindle (±s.e.m) as a function of egg length in wild-type and *ect-2(xs110)* embryos. (**F**) Quantification of the mean position of furrow initiation (±s.e.m) relative to the center of the spindle midzone in wild-type and *ect-2(xs110)* embryos. (***p < 0.001, by t-test).

The following figure supplements are available for figure 6:

**Figure supplement 1**. Viability and fertility of *cyk-4* mutants and suppressors.

**Figure supplement 2**. Conservation of ECT-2 GEF domain and inferred position of the *ect-2(xs110)* allele.

**Figure supplement 3**. Accumulation of myosin during cytokinesis in *ect-2(xs110)*.

**Figure supplement 4**. *ect-2(xs110)* and *ect-2(xs111)* are dominant gain of function mutations.

The animals were maintained at the permissive temperature for two generations before shifting to the restrictive temperature. We were able to isolate a strain that was viable and fertile. The *ect-2* locus was sequenced and de novo generation of the G707D substitution was confirmed. This mutation in *ect-2* therefore suppresses all the essential functions affected by the *cyk-4(or749ts)* allele.

We next investigated whether this mutation causes a detectable phenotype when separated from *cyk-4(or749ts)*. Interestingly, *ect-2(xs110)* embryos exhibit hypercontractility during both pseudo-cleavage and cytokinesis (*Figure 6Bi*); this hypercontractility is associated with enhanced cortical accumulation of myosin II (*Figure 6—figure supplement 3*). Hypercontractility is also observed in embryos from *ect-2(xs110)/+* hermaphrodites, indicating *ect-2(xs110)* is a dominant, gain of function allele (*Figure 6—figure supplement 4*). The hypercontractility is reduced in *ect-2(xs110); cyk-4 (or749ts)* embryos (ii), indicating that ECT-2$^{G707D}$ hyperactivity is partially dependent on CYK-4 and that *cyk-4(or749ts)* and *ect-2(xs110)* exhibit mutual suppression. Comparison of *ect-2(xs110); cyk-4 (or749ts)* embryos to *ect-2(xs110); cyk-4(or749ts); nop-1(RNAi)* (*Figure 6ii* vs *Figure 6iii*) embryos reveals that NOP-1 also contributes to contractility in ECT-2$^{G707D}$ embryos.

An unusual phenotype was observed in *ect-2(xs110)* embryos. Following anaphase, the cleavage furrow frequently initiates from a site significantly anterior to the midpoint of the anaphase spindle (*Figure 6D–F*). As the furrow ingresses, it undergoes a dramatic repositioning so that it ultimately bisects the anaphase spindle. Nevertheless, the *ect-2(xs110)* strain is viable and fertile despite exhibiting hypercontractility during polarization and cytokinesis (*Figure 6—figure supplement 1*).

The second suppressor allele, *ect-2(xs111)*, also contains a substitution mutation in ECT-2. This mutation is located in the linker region between the cryptic BRCT0 domain and BRCT1 (*Zou et al., 2014*) (*Figure 6A*). Several criteria indicate that this mutation is also causal. First, SNP mapping placed suppressor activity near the *ect-2* locus. Second, the suppressor was analyzed by one step mapping and whole genome sequencing (*Doitsidou et al., 2010*). *ect-2* is the only gene in the candidate region that contained a non-silent mutation that has any role in cytokinesis. Third, biochemical data indicate that this mutation relieves ECT-2 autoinhibition (see below). The *ect-2(xs111)* gain of function allele exhibited similar overall characteristics as *ect-2(xs110)* (*Figure 6B,C*), although the spindle positioning defect was less severe (not shown). The one remarkable difference was that *ect-2(xs111); cyk-4 (or749ts); nop-1(RNAi)* (*Figure 6Bviii*) embryos fully ingressed during cytokinesis, though they do not form pseudocleavage furrows. The ability of these embryos to complete cytokinesis depends upon residual activity from CYK-4$^{E448K}$, as depletion of CYK-4 by RNAi prevents completion of cytokinesis in *ect-2(xs111)* embryos (*Figure 6Bx*). As complete furrow ingression is not seen in comparable *ect-2 (xs110)* embryos (*Figure 6Biii*), *ect-2(xs111)* may be more strongly activated than *ect-2(xs110)*.

This genetic screen demonstrates that only rare mutations suppress *cyk-4(or749ts)* and that the essential function of CYK-4 that is inactivated by CYK-4$^{E448K}$ is the ability to activate RhoA. Note that while *ced-10(n1993)* can partially suppress cytokinesis defects in CYK-4$^{E448K}$ expressing embryos (cytokinesis remains delayed and slow in the double mutant; and only ~67% of embryos complete division), *ced-10(n1993)* does not rescue *cyk-4(or749ts)* to viability (*Figure 6—figure supplement 1*).

CYK-4$^{R459A}$ causes a less severe phenotype than CYK-4$^{E448K}$, therefore we predicted that *ect-2 (xs110)* and *ect-2(xs111)* could also suppress CYK-4$^{R459A}$. We used CRISPR/Cas9 to introduce the R459A mutation into the endogenous *cyk-4* gene and crossed it into both *ect-2* hyperactive mutants. We were able to isolate strains in which the sole source of CYK-4 lacks the critical arginine in the active site (*Figure 6Biv,ix*). The resulting strains exhibited high viability and fertility (*Figure 6—figure supplement 1*). This finding provides independent confirmation that the sole essential function of the RhoGAP active site of CYK-4 is to stimulate ECT-2-mediated RhoA activation.

### *cyk-4* suppressor mutations activate *ect-2*

These genetic and cell biological results demonstrate that the GAP activity of CYK-4 contributes to RhoA activation. As ECT-2 is required for all RhoA activity during cytokinesis, the CYK-4 GAP domain is likely to serve this role by modulating ECT-2. Given that the canonical function of a RhoGAP domain is to inhibit RhoA activity, it is surprising that a protein containing a RhoGAP domain enhances RhoA activation. However, CYK-4 and ECT-2 form a protein complex through their regulatory N-termini (*Burkard et al., 2007*; *Wolfe et al., 2009*), therefore the C-terminal GAP domain of CYK-4 will be in the vicinity of the ECT-2 RhoGEF domain.

We therefore hypothesized that the interactions between CYK-4 and ECT-2 are not limited to their N-termini. To test this, we purified the C-terminal domains of CYK-4 and ECT-2 (*Figure 7A*) and performed binding assays. We found that the catalytic C-termini of CYK-4 and ECT-2 directly interact (*Figure 7B*, *Figure 7—figure supplement 1*); a similar complex is also found with human orthologs (data not shown). We assayed for activation of the ECT-2 GEF activity by the CYK-4 GAP domain in vitro. However, we have not yet been able to detect stimulation of GEF activity (data not shown). This negative result could be due to missing components, a requirement for the context provided by the full length, oligomerization competent proteins (*Basant et al., 2015*), or the absence of the plasma membrane to which CYK-4 must bind in vivo in order to activate ECT-2.

As *ect-2(xs110)* and *ect-2(xs111)* suppress the phenotypes caused by mutations in the CYK-4 GAP domain, we sought to understand the biochemical basis of activation by the proteins they encode, ECT-2$^{G707D}$ and ECT-2$^{E129K}$, respectively. Given that the E129K mutation lies near the N-terminal BRCT domain (*Figure 7A*), we hypothesized that it could interfere with ECT-2 autoinhibition. To test this possibility, we assayed for binding between the N- and C- termini of ECT-2. Wild-type N- and C-termini form a complex that is readily detected in vitro. However, the E129K substitution, but not G707D, significantly reduces binding of the ECT-2 N- and C-termini (*Figure 7C*, *Figure 7—figure supplement 1*), suggesting that this allele functions by relieving autoinhibition.

The G707D mutation in ECT-2 is located in the PH portion of the RhoGEF domain (*Figure 7A*, *Figure 6—figure supplement 2*). In principle, this mutation could promote RhoA activation by a number of mechanisms including activation of the GEF domain, relieving autoinhibition of ECT-2, stabilizing the interaction with CYK-4 and/or stabilizing the interaction of ECT-2 with the plasma membrane. We did not observe a change in the association of ECT-2_N with ECT-2_C$^{G707D}$, suggesting that the mutation doesn't relieve autoinhibition. However, as the mutated residue maps to a helix that lies near RhoA in a co–crystal structure of a related RhoGEF (*Figure 6—figure supplement 2*), we tested whether it activates RhoGEF activity. We assayed ECT-2 GEF activity in an in vitro exchange assay. The ECT-2$^{G707D}$ variant exhibits a modest increase in GEF activity compared to wild-type ECT-2 over a range of concentrations (*Figure 7D*), perhaps by increasing the affinity of ECT-2 for RhoA.

## Discussion

### Overview

Diverse mechanisms ensure that the cytokinetic contractile ring assembles at the cell equator following chromosome segregation. These regulatory mechanisms converge to promote local accumulation of active RhoA at the cell equator which is an essential prerequisite for contractile ring assembly. Whereas it is widely accepted that the RhoGEF ECT-2 is the primary activator of RhoA and the centralspindlin component CYK-4 contributes to RhoA activation, the mechanism(s) by which CYK-4 promotes RhoA activation have been rather unclear. Here, we demonstrate that CYK-4 has multiple functional domains that are required for it to promote RhoA activation. In addition to the previously characterized binding interaction with ECT-2, we show that both the C1 domain and the

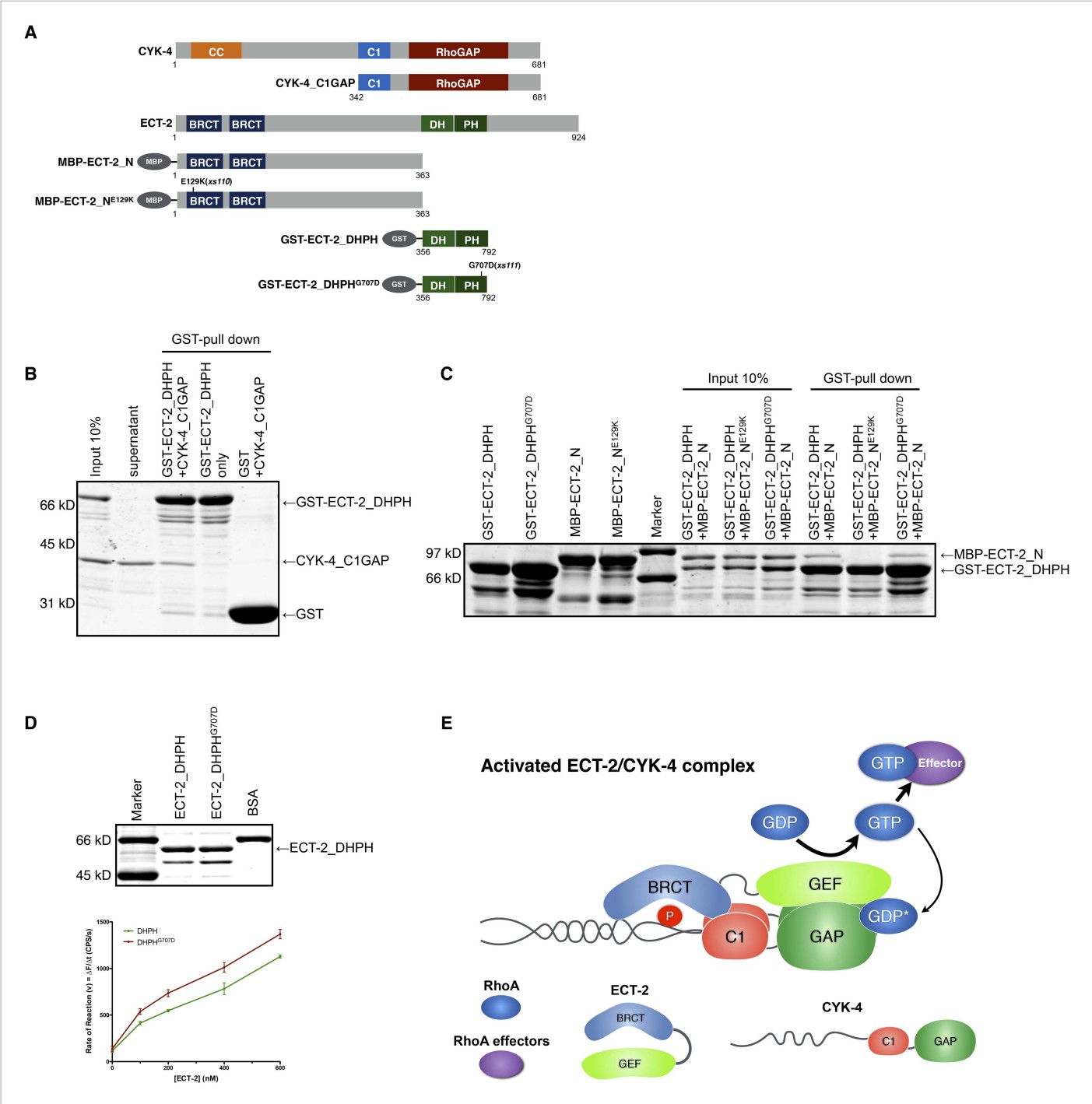

**Figure 7**. Biochemical basis for suppression by ECT-2 variants. (**A**) Schematic depiction of the domain organization of CYK-4 and ECT-2 and the recombinant fragments used for biochemical analyses. (**B**) GST pulldown between GST-ECT-2 DHPH and CYK-4 C1GAP. (**C**) Pulldown assay between MBP-ECT-2-N (wild-type and the E129K variant) with GST-ECT-2 DHPH (both wild-type and the G707D variant). The MBP proteins were present in the soluble fraction and incubated with the GST-DHPH fragments bound to beads. The wild-type N-terminus associates with wild-type and G707D C-termini. However, the ECT-2 N[E129K] is defective in binding to wild-type C-terminus. (**D**) The G707D substitution activates the exchange activity of ECT-2. Exchange assays were performed with RhoA•GDP exchanging for mant-GTP at different concentrations of ECT-2 DHPH and ECT-2 DHPH[G707D]. Results shown are the average (±s.e.m) of three assays. The gel contains 1 μg of each ECT-2 variant and a BSA standard. (**E**) Working model summarizing the proposed mechanism for ECT-2 activation. Note that only the CYK-4 subunit of centralspindlin is shown. In vivo, centralspindlin is predicted to be oligomeric and the entire complex bound to the plasma membrane.

*Figure 7. continued on next page*

*Figure 7. Continued*

The following figure supplements are available for figure 7:

**Figure supplement 1**. Biochemical characterization of CYK-4 and ECT-2 variants.

**Figure supplement 2**. Proposed states of the ECT-2/CYK-4 complex.

catalytic activity of the RhoGAP domain of CYK-4 are also required for full activation of ECT-2. Furthermore, our results indicate in order for the CYK-4 GAP domain to promote RhoA activation, it has to act catalytically on RhoA•GTP. This implies that RhoA plays a role in promoting its own activation.

## The GAP activity of CYK-4 is required for RhoA activation

Four results suggest that CYK-4 GAP activity promotes RhoA activation. First, CYK-4 GAP activity is required for the completion of cytokinesis and embryos lacking this activity exhibit reduced levels of RhoA effectors. Second, when the NOP-1-dependent, parallel pathway for RhoA activation is eliminated, the requirement for CYK-4 GAP activity for furrow formation and effector recruitment is greatly enhanced. Third, we have demonstrated a biochemical interaction between the GAP domain of CYK-4 and the GEF domain of ECT-2. Fourth, the requirement for CYK-4 GAP activity can be alleviated by three independent perturbations that each increase RhoA activity levels.

One of the perturbations that suppresses the CYK-4 GAP-deficient phenotype is depletion of the primary RhoA GAP, RGA-3/4. Embryos defective in RGA-3/4 alone exhibit hypercontractility and are largely inviable; these phenotypes are consistent with the canonical function of a RhoA GAP. However, loss of CYK-4 GAP activity and loss of RGA-3/4 counterbalance each other during cytokinesis. Furthermore, suppression by RGA-3/4 depletion is potent, it can restore cytokinesis in embryos deficient in both NOP-1 and CYK-4 GAP activity in which furrows otherwise barely ingress.

A large, unbiased, genome-wide screen for suppressors of *cyk-4(or749ts)* corroborates the model that CYK-4 GAP activity promotes RhoA activation. We identified two strong, extragenic, gain of function suppressor mutations in the RhoGEF ECT-2 (*Figure 6*). Because these suppressors rescue *cyk-4(or749ts)* and *cyk-4$^{R459A}$* to viability, the essential function of the CYK-4 GAP activity must be to promote RhoA activation.

## Mechanism of ECT-2 activation and positive feedback

These results raise the fundamental question: by what mechanism does the GAP activity of CYK-4 contribute to RhoA activation? We propose a working model in which the most active form of the ECT-2 RhoGEF is a complex containing ECT-2 and CYK-4 with a molecule of RhoA•GDP bound to the GAP active site (*Figure 7E*).

The GTPase bound to CYK-4 is likely to be RhoA, rather than CED-10/Rac1 or CDC-42. If CYK-4 had to bind CED-10/Rac1 or CDC-42, then depletion of those GTPases should impair cytokinesis as severely as a mutation that attenuates GTPase binding by the CYK-4 GAP domain. However, neither CED-10/Rac1 nor CDC-42 is required for cytokinesis, even in NOP-1-defective embryos (*Figure 3*), whereas weakening GTPase binding by the CYK-4 GAP domain strongly impacts cytokinesis. The model has a further implication: to form the most active ECT-2 GEF complex, CYK-4 GAP binds RhoA•GTP. Therefore, RhoA•GTP plays a role in RhoGEF activation, suggesting the presence of a positive feedback loop during cytokinesis.

This working model is supported by the finding that full activation of RhoA and cytokinesis requires that the CYK-4 GAP domain both bind a Rho family GTPase (*Figure 3*) and activate its ability to hydrolyze GTP (*Figures 2, 4, 5*); indeed mutations in the CYK-4 GAP domain that diminish GTPase binding exhibit a stronger defect in RhoA activation than mutation of the GAP active site (*Figure 5*). Our mutational analysis has trapped ECT-2 in four distinct states that form an allelic series (*Figure 7—figure supplement 2A*). We propose that the least active form of ECT-2 is not bound to CYK-4 and has little GEF activity. Once CYK-4 is phosphorylated, it can be bound by the ECT-2 N-terminal BRCT domains rendering it weakly activated (equivalent to CYK-4$^{EE}$). This form may also

exhibit some interactions between the GEF domain of ECT-2 and the GAP domain of CYK-4, as these domains can interact in vitro without RhoA present. If CYK-4 can bind to RhoA•GTP, it induces a higher activity state, as evidenced by the increased activity of CYK-4$^{R459A}$, which is sufficient for zygotic development. Finally, if CYK-4 can induce GTP hydrolysis by RhoA, this results in the fully active ECT-2/CYK-4/RhoA•GDP or ECT-2/CYK-4/RhoA•GDP + Pi state populated by the wild-type protein. We speculate that this complex results in full relief from autoinhibition within the ECT-2 GEF domain.

Not only does this model explain why CYK-4 retains GAP activity towards RhoA, it also explains why its ability to inactivate RhoA is attenuated relative to Rac and Cdc42. High turnover rates of RhoA•GTP induced by CYK-4 might rapidly consume RhoA•GTP at the site of production, yielding a futile cycle of RhoA activation and inactivation. However, this working model must be tested by structural studies and biochemical reconstitution assays that reflect the in vivo situation. Accurate reconstitutions will need to account for the facts that cytokinetic RhoA activation involves the CYK-4 C1 domain (*Figure 1C*), the ability of CYK-4 to bind to ZEN-4, and the ability of ZEN-4 to oligomerize (*Basant et al., 2015*).

## CED-10/Rac1 is not the primary target for CYK-4 GAP activity

We tested the model that GAP activity of CYK-4 is important to maintain CED-10/Rac1 in an inactive state. Some of our results do support this model, as the failure to complete the first cytokinesis in embryos lacking CYK-4 GAP activity can be partially restored by a loss of function mutation in *ced-10/Rac1* (*Figure 4B*). We therefore tested whether inactivation of CED-10/Rac1 suppresses loss of CYK-4 GAP activity in NOP-1-deficient embryos. GAP-deficient CYK-4 does not promote significant ingression of the cleavage furrow in embryos lacking NOP-1, irrespective of the presence or absence of CED-10/Rac1 (*Figure 4Biii,v*). Finally, mutations in CED-10/Rac1 do not suppress the lethality of a temperature sensitive mutation in *cyk-4* (*Figure 6—figure supplement 1*).

Thus, because the active site of the CYK-4 GAP domain is required in the absence of CED-10/Rac1, CED-10/Rac1 inactivation cannot be the primary function of the CYK-4 GAP domain. Rather, these results suggest a model in which loss of CED-10/Rac1 function causes a reduction in overall cortical tension which, in turn, allows an increase in the extent of NOP-1-dependent furrow ingression (*Loria et al., 2012*).

## GEF activation model can account for many previous results

The experiments presented here demonstrate that CYK-4 GAP activity promotes RhoA activation and that this function is essential in early *C. elegans* embryos and in the adult germline (*Figures 2A, 4B*). Our experiments also addressed the function of CYK-4 GAP activity post-embryonically. We find that whereas zygotic *cyk-4* null embryos die during embryogenesis, expression of catalytically inactive CYK-4 provides significant rescue, supporting development into viable, albeit sterile, adults (*Figure 2A*). Thus, while early embryos require the GAP activity of CYK-4, this requirement is relaxed post-embryonically. The requirement for CYK-4 GAP activity can be experimentally eliminated by hyperactivation of ECT-2 or depletion of the RhoA GAP RGA-3/4 (*Figures 5B, 6B*). Interestingly, RGA-3/4 is primarily expressed in the germline and in early embryos (NextDB, cited in *Schmutz et al., 2007* and data not shown), thus regulated RGA-3/4 expression could contribute to the tissue specific requirements.

These findings allow us to reconcile many previous results on the role of the CYK-4 GAP domain during cytokinesis. Some studies provided evidence that the GAP activity is dispensable (*Goldstein et al., 2005*; *Yamada et al., 2006*), whereas others suggested it is required for Rac1 inactivation (*D'Avino et al., 2004*; *Canman et al., 2008*; *Bastos et al., 2012*), RhoA inactivation (*Miller and Bement, 2009*), or RhoA activation (*D'Avino et al., 2004*; *Zavortink et al., 2005*; *Loria et al., 2012*). The first set of results is consistent with the results presented here, as some cell types may not require the GAP activity of CYK-4 for cytokinesis, as seen in post embryonic cells in *C. elegans*. As numerous studies have shown that active RhoA can indirectly inhibit Rac1 (see *Guilluy et al., 2011* for review), some of the results that point to a role for CYK-4 GAP activity in attenuating Rac1 levels (*Bastos et al., 2012*) may be indirectly caused by a reduction RhoA activation or by indirectly controlling cortical tension. Thus, many previous results can be explained without proposing that the CYK-4 GAP domain performs different functions in different organisms or cell types. We do not rule out the possibility

that, in certain contexts, CYK-4 or its orthologs negatively regulate Rac or Cdc42. Recent evidence indicates that the *Xenopus* ortholog of CYK-4 concentrates at cell–cell junctions and negatively regulates GTPases at that site (*Breznau et al., 2015*). Further work is required to resolve why CYK-4 acts as a positive regulator of RhoA in *C. elegans* embryos and a negative regulator in *Xenopus* embryos (*Miller and Bement, 2009*; *Breznau et al., 2015*).

### Regulation of RhoGEFs by GTPases and GAPs

The signaling mechanisms we have discovered in cytokinesis have analogies in other signaling pathways. Our favored model, in which RhoA promotes its own activation, is reminiscent of the positive feedback in Cdc42 activation during yeast budding (*Howell and Lew, 2012*) and the activation of the SOS1 RasGEF domain by a molecule of Ras•GTP that serves as an allosteric activator (*Gureasko et al., 2008*). Interestingly, ECT-2 has also been implicated in SOS regulation (*Canevascini et al., 2005*). Likewise, CYK-4 is not the only protein with a GTPase activating domain that plays a role in promoting GEF activity. A similar function, in cis, has been seen in p115 RhoGEF, which is activated by Gα13 (*Chen et al., 2012*). In this case, the binding of a molecule of Gα13 to an allosteric site on the RhoGEF domain of p115 is stabilized by p115's N-terminal RGS domain (*Figure 7—figure supplement 2B*). RGS domains accelerate GTP hydrolysis by Gα, that is, they are Gα GAPs (*Tesmer et al., 1997*). As RhoGEF activation is not an obvious function for a RhoGAP domain, additional cases may have gone undetected.

### Complex control of RhoA activation during cytokinesis

RhoA activation is controlled by multiple layers of regulation during cytokinesis. In addition to cell-cycle regulated changes in the phosphorylation state of CYK-4 and ECT-2 that control their binding and localization (*Yüce et al., 2005*; *Su et al., 2011*; *Zou et al., 2014*), full activation of RhoA also involves membrane binding by CYK-4 (*Figure 1*) which, in turn, requires centralspindlin oligomerization (*Basant et al., 2015*). Like the requirement for CYK-4 GAP activity, the requirement for the C1 domain of CYK-4 is context dependent. Whereas the C1 domain makes a major contribution to furrow ingression in *C. elegans* embryos, studies in Hela cells demonstrate that the C1 domain contributes to RhoA activation, but it is not essential (*Lekomtsev et al., 2012*). Thus, centralspindlin has several domains that contribute to maximal activation of ECT-2. However, not all cell types may require maximal activation of ECT-2 either because of physical properties of the cell (cell size, cortical tension, and tissue organization) or because of their biochemical properties (expression of RGA-3/4 orthologs). Nevertheless, the evolutionary conservation of all of these functions suggests that they play critical roles during some stage(s) of metazoan development.

## Materials and methods

### *C. elegans* strains

Animals were grown at 20°C on standard nematode growth media (NGM) plates seeded with OP50 *Escherichia coli*. Some strains were provided by the Caenorhabditis Genetics Center. All strains used in this study are listed in *Supplementary file 1*.

### RNAi

RNAi was administered by feeding nematodes with *E. coli* expressing the appropriate double-stranded RNA (dsRNA) (*Timmons and Fire, 1998*). HT115 bacterial cultures were grown in Luria broth with 100 μg/ml ampicillin overnight at 37°C. Cultures (250 μl) were seeded on NGM plates containing 100 μg/ml ampicillin and 1 mM IPTG and incubated at room temperature for 16 hr. RNAi plasmids were obtained from the library produced by *Kamath et al. (2003)*. Young L4 hermaphrodites were picked onto the plates for feeding at 25°C at least 24 hr prior to dissection. For RNAi depletion of temperature-sensitive alleles, L4 larvae were fed for 48 hr at 16°C, then shifted to 25°C for at least 1 hr before imaging.

For experiments where two genes were simultaneously knocked down by RNAi, bacterial cultures of *E. coli* expressing the appropriate dsRNA were mixed in a 1:1 ratio seeded onto NGM plates as described above. If stronger depletion of one of the two genes was desired, embryos were first

hatched onto feeding plates targeting the gene. L4 worms were transferred to fresh plates with bacteria expressing dsRNA against both genes.

## Egg hatch assay

Young gravid hermaphrodites were transferred to fresh seeded NGM plates in triplicate. Remove worms from plates after ~8 hr of egg laying. The eggs laid on plates were scored manually under dissecting microscope. To determine unhatched embryos, embryos remaining on plates were scored 1 day after the parents were removed. The embryonic lethality percentage is calculated as the number of unhatched embryos divided by the total egg production.

## Plasmid construction

To generate CYK-4::GFP MosSCI constructs, ~2 kb sequences upstream of *cyk-4*, *cyk-4* genomic DNA tagged with C-terminal GFP coding sequences, and *pie-1* 3′ UTR sequences were generated by overlapping PCR and inserted to pCFJ150 by SLiCE (*Zhang et al., 2012*). *cyk-4* genomic sequences between BamHI and AvrII were recoded to generated RNAi resistant alleles. To introduce *cyk-4* mutations, sequences covering mutations were generated by overlapping PCR using pCJF150-cyk-4-gfp as template and the appropriate primers (see primer sequences in *Supplementary file 2*): MG4199/ MG4276 and MG4200/MG4277 for E448K; MG4199/MG4202 and MG4200/MG4201 for R459A; MG4199/MG4489 and MG4200/MG4488 for K459E/R499E(EE); MG4199/MG4070 and MG4200/ MG4071 for ΔC1. Overlapping PCR products were inserted into pCFJ15-cyk-4-gfp linearized with NaeI by SLiCE. All constructs were sequence verified.

Cas9/sgRNA plasmids were derived from pDD162 vector (*Dickinson et al., 2013*). *ect-2* sgRNA target sequences were generated by overlapping PCR using pDD162 as PCR template and the appropriate primers (see primer sequences in *Supplementary file 2*), MG4735/MG4773 and MG4774/ MG4736. Overlapping PCR products were inserted into pDD162 linearized with SpeI/BsrBI by SLiCE.

## Transgenes and germline transformation

Transgenic lines expressing single copy CYK-4::GFP or mutant CYK-4::GFP were generated by integrating constructs into the Mos1 element ttTi5605 on chromosome II using the MosSCI method (*Frøkjaer-Jensen et al., 2008*).

For oligonucleotide templates (ODNs) based CRISPR experiments (*Zhang and Glotzer, 2014*; *Zhao et al., 2014*), microinjection was performed by injecting DNA mixture into gonad arms of *cyk-4 (or749ts)* young gravid hermaphrodites. Injected *cyk-4(or749ts)* were maintained at 16°C for 3–4 days then shifted to 25°C until starvation. Viable worms were isolated and subjected to single worm PCR to identify desired mutations. The injection mixture consists of Cas9/sgRNA plasmids and ODNs. The final concentrations of plasmids and ODNs are Cas9/ect-2 sgRNA vector at 50 ng/μl and *ect-2* ODN (MG4801 5′-TTGTATGGTGCCTGATTCATCGTGACGAGCAAGATGGTGACATTGACACAGTC TTCGAAT-3′) at 50 ng/μl.

## Microscopy

To prepare one-cell embryos for imaging, gravid hermaphrodites were dissected into egg salt buffer (HEPES pH 7.4 5 mM, NaCl 118 mM, KCl 40 mM, MgCl$_2$ 3.4 mM, CaCl$_2$ 3.4 mM) on coverslips, mounted onto 2.5% agar pads and sealed with vaseline. For Nomarski imaging, embryos were observed with a Zeiss (Thornwood, NY) Axioplan II with a 100×/1.3 Plan-Neofluar objective. Images were captured with a charge-coupled device (CCD) camera (Imaging Source, Charlotte, NC) controlled by Gawker (gawker.sourceforge.net). Images were acquired every 5 s and processed with ImageJ (http://rsbweb.nih.gov/ij). For confocal imaging, embryos were imaged with a 63×/1.4 oil-immersion lens on (1) a Zeiss Axiovert 200M equipped with a Yokogawa CSU-10 spinning-disk unit (McBain, Simi Valley, CA) and illuminated with 50-mW, 473-nm and 20-mW, 561-nm lasers (Cobolt, Solna, Sweden), or (2) a Zeiss Axioimager M1 equipped with a Yokogawa CSU-X1 spinning-disk unit (Solamere, Salt Lake City, UT) and illuminated with 50-mW, 488-nm and 50-mW, 561-nm lasers (Coherent, Santa Clara, CA). Images were captured on a Cascade 1K EM-CCD camera or a Cascade 512BT (Photometrics, Tucson, AZ) controlled by MetaMorph (Molecular Devices, Sunnyvale, CA). Image processing was performed with ImageJ. Time-lapse acquisitions were assembled into movies using Metamorph and ImageJ.

## Image quantification

To measure furrow ingression kinetics, a single central plane image of GFP::PH or mCherry::PH was acquired at 10 s intervals starting at anaphase as assessed by the CYK-4::GFP or mCherry::HIS-58 signal. The position of the furrow was assessed in each frame by manual tracking of GFP::PH or mCherry::PH signal. The extent of ingression in each frame was calculated as d/w, where w is the total width of the embryo and d is the distance between the furrow tips. To determine whether furrow ingression kinetics were statistically significant different between multiple genotypes, data sets of normalized cortical distance from 100 s to 410 s after anaphase onset were analyzed with a Kruskal–Wallis non-parametric one-way analysis of variance (ANOVA) using Dunnett's multiple comparisons test.

To quantitate the abundance of NMY-2::mRFP at the equatorial region, a stack of five planes spanning 2.5 μm was captured every 10 s. The Z-stacks were projected using a maximum intensity projection algorithm and corrected for photobleaching. Using custom ImageJ macros, the background signal was measured in a remote region of each frame. A region of fixed size in the equatorial region was thresholded with a minimal value of 1.25× background and the total thresholded signal in the region was integrated and normalized to the background. The total value of intensity was summed for a defined number of planes after anaphase onset.

To quantitate the abundance of CYK-4::GFP at the furrow tip, a stack of five planes spanning 2.5 μm was captured every 10 s. The frame in which furrow ingressed to half of the egg width or the deepest was chosen. The Z-stacks were projected using a maximum intensity projection algorithm and corrected for photobleaching. The background was measured as integrated intensity of a square adjacent to the furrow tip. The GFP intensity was measured as integrated intensity of a square with the same area covering the furrow tip and normalized to the background by subtracting background intensity (see *Figure 1*).

## Protein expression and purification

The coding sequences for CYK-4 C1 and RhoGAP domain (342-681aa), ECT-2 DH/PH domain (356-792aa), CED-10/Rac1 and RhoA (*C.e.*) were cloned into the GST expression vector pGEX-4T-tobacco etch virus (TEV), and the coding sequences for ECT-2 BRCT domain (1-363aa) were cloned into MBP expression vector pMAL-c2-TEV. GST- and MBP-tagged proteins were expressed in *E. coli* strain BL21 by adding 0.3 mM IPTG at OD600 reached 0.5–0.7 at 25°C. Cells were grown for another 4 hr at 25°C and collected. Frozen cells were thawed in lysis buffer (50 mM Tris pH 7.5, 150 mM NaCl, 5 mM $MgCl_2$, 10% Glycerol, 1 mM PMSF, 1 μg/ml leupeptin, 1 μg/ml pepstatin A, 0.1% Triton X-100, 1 mM DTT, 0.5 mg/ml lysozyme) and lysed by sonication. The bacterial lysate was centrifuged at 40,000×*g* at 4°C for 30 min.

For GST-tagged proteins, glutathione-Sepharose 4B beads (bioWORLD) were added to supernatant and incubated at 4°C for 4 hr. The beads were washed 3× with 50 mM Tris pH 7.5, 150 mM NaCl, 5 mM $MgCl_2$, 1 mM DTT. Protein-bound beads in were either stored in 50 mM Tris pH 7.5, 150 mM NaCl, 5 mM $MgCl_2$, 1 mM DTT, 50% glycerol at −20°C, or cleaved from beads by incubation with His-tagged TEV protease at 4°C overnight. TEV protease was removed by incubation with TALON beads (Clontech). Cleaved fusion proteins were stored in 10% glycerol at −80°C.

For MBP-tagged proteins, amylose resin (New England Biolabs) was added to supernatant and incubated at 4°C for 4 hr. Beads were placed in a poly-prep chromatography column (Bio-Rad), and washed with 12 column volumes of 50 mM Tris pH 7.5, 150 mM NaCl, 5 mM $MgCl_2$, and 1 mM DTT. Fusion proteins were eluted with 50 mM Tris pH 7.5, 150 mM NaCl, 5 mM $MgCl_2$, 1 mM DTT, 10 mM maltose, and stored in 10% glycerol at −80°C.

## GEF activity measurements

*C.e.* GST-RhoA was loaded with GDP (*Self and Hall, 1995*). Beads were washed with low magnesium buffer (50 mM Tris pH 7.5, 150 mM NaCl, 1 mM EDTA, 1 mM DTT) and 1 mM GDP was added. Beads were incubated with shaking at room temperature for 15 min, placed on ice, and 20 mM $MgCl_2$ was added and incubated on ice for 5 min. Beads were washed three times with 50 mM Tris pH 7.5, 150 mM NaCl, 5 mM $MgCl_2$, 1 mM DTT. GDP-loaded RhoA was cleaved from beads with TEV protease, glycerol added to 10%, flash frozen and stored at −80°C.

Fluorescence-based kinetic assays were performed in HORIBA FluoroLog-3 Spectrofluorometer, with fluorescence analog of GTP, mant-GTP (AnaSpec). All nucleotide exchange assays were

performed in the presence of 1 µM RhoA-GDP, 200 nM mant-GTP, the indicated concentration of ECT-2 DH/PH domain in 20 mM Tris pH 7.5, 50 mM NaCl, 10 mM $MgCl_2$, 1 mM DTT, 50 µg/ml BSA, 1% glycerol. The relative fluorescence was monitored for 90 s before adding mant-GTP, and for 510 s after adding mant-GTP; measurements were taken every 15 s. The reaction rate, v, is defined as $\Delta F/\Delta t$, where F = fluorescence, t = time.

### RhoGAP assay

CYK-4 GAP (final concentration from 0 to 800 nM) and RhoA or Rac1 (final concentration 9 µM) were mixed in 1× reaction buffer (50 mM Tris pH 7.5, 50 mM NaCl, 5 mM $MgCl_2$, 1 mM DTT, 1% glycerol), then GTP was added to 1 mM to start the reaction. After 30 min, inorganic phosphate was assayed using a malachite green-based assay (Kodama et al., 1986); absorbance was measured with a NanoDrop 2000 spectrophotometer (Thermo Scientific). For time course experiments, 200 nM CYK-4 GAP was added into the reaction and, at the indicated time points, aliquots of the reaction were removed to assess free phosphate.

### GST pull-down assay

For each binding experiment, purified fusion proteins were added to the protein-bound glutathione-sepharose beads and incubated for 1 hr at 4˚C. After three washes in cold wash buffer (50 mM Tris pH 7.5, 150 mM NaCl, 5 mM $MgCl_2$, 1 mM DTT), proteins were eluted into loading buffer, separated by SDS-PAGE, and detected by coomassie blue staining.

## Acknowledgements

This work was supported by the NIGMS (GM085087), a Chicago Postdoctoral Fellowship to DZ, and NCATS (UL1 TR000430). We thank Benjamin Wolfe, Andy Loria, Agnieszka Grzegorzewska, and Yael Feinstein for their contributions to early stages of this project. We also thank Bruce Bowerman, Douglas Bishop, and Angika Basant for helpful comments on the manuscript. We acknowledge the support of the *C. elegans* stock center which provided some strains.

## Additional information

### Funding

| Funder | Grant reference | Author |
| --- | --- | --- |
| National Institutes of Health | GM085087 | Michael Glotzer |
| University of Chicago | | Donglei Zhang |
| National Institutes of Health | UL1 TR000430 | Michael Glotzer |

The funders had no role in study design, data collection and interpretation, or the decision to submit the work for publication.

### Author contributions

DZ, Conception and design, Acquisition of data, Analysis and interpretation of data, Drafting or revising the article; MG, Conception and design, Analysis and interpretation of data, Drafting or revising the article

### Author ORCIDs

Michael Glotzer, http://orcid.org/0000-0002-8723-7232

## Additional files

### Supplementary files

• Supplementary file 1. *C. elegans* strains used in this study.

• Supplementary file 2. List of oligonucleotides used in this study.

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
