## [Decision Letter]

Thank you for submitting your work entitled “The RhoGAP activity of CYK-4/MgcRacGAP functions non-canonically by promoting RhoA activation during cytokinesis” for peer review at *eLife*. Your submission has been favorably evaluated by Randy Schekman (Senior Editor) and three reviewers, one of whom, Mohan Balasubramanian, is a member of our Board of Reviewing Editors.

The reviewers have discussed their reviews with one another, and the Reviewing Editor has drafted this decision to help you prepare a revised submission.

As you see, the views were varied (but all supportive), and Reviewer 3 has raised a number of technical issues all of which seem reasonable and I would like you to consider and respond to these comments. In particular: 1) please provide in vivo data to address key points of Reviewer 3 (such as potential effect of remaining Rac activities; RAC-2 or MIG-2), and 2) please provide proper controls for in vitro experiments raised by Reviewer 3.

As for Reviewer 2, although the biochemical experiments are ultimately required to advance your model further, this may not be accomplished in a single paper and may not be possible within the timeframe of the revision. If you have made some progress since submission of your paper and if they address the mechanism of GAP activation of ECT2, you may add it.

Please also address the points of Reviewer 1, most of which seem to be aimed at improving the readability as well as placing the paper in the right context through further discussion. Please see below the comments of the three reviewers.

Reviewer #1:

This is an interesting paper by Glotzer and colleagues in which they have investigated the function of CYK-4 GAP protein in *C. elegans* cytokinesis. Through structure-function analysis of CYK-4 GAP they have identified that the GAP activity of CYK-4 is essential for cytokinesis, and performs at least 2 different functions. They make the important discovery that the GAP activity is crucial for nucleotide exchange on RhoA, the GTPase that activates both actin polymerization and myosin II activation. The work is very thorough and I particularly like the experiment where, through an unbiased screen for suppressors of CYK-4 GAP mutants, they identify new alleles of the RhoA exchange factor ECT2 itself. They further go on to show that CYK-4 physically interacts with ECT-2 and that an Ect2 mutant they isolated has higher nucleotide exchange activity (albeit a modest one).

I think the paper is appropriate for publication in *eLife*. However, I would like to see the biochemistry developed better. In particular, I am surprised that none of the GAP mutants identified in the work have been characterized in the biochemical studies described in the latter part of the paper. The two key experiments are: 1) how do the GAP mutants behave in the physical interaction experiments with ECT2 and 2) how does the presence of wild-type or GAP mutants of CYK-4 affects the ECT2-mediated nucleotide exchange on RhoA. These experiments are crucial if they are to advance the model in Figure 7.

I think some more discussion is also warranted on how mutations in the PH domain potentially affect the GEF activity. Molecular dynamics may help, although I realize this can be a new foray altogether.

Reviewer #2:

Zhang and Glotzer provide an extensive and compelling analysis of the controversial role of the *C. elegans* RhoGAP CYK-4 during cytokinesis in animal cells. Their data strongly support the conclusion that the primary role of CYK-4/RhoGAP is to activate RhoA through its RhoGEF ECT-2. This is a counter-intuitive finding, as normally one thinks of GAPs as negative regulators of GTPases, not activators. The authors provide a rationale for this surprising role, and their evidence certainly supports the conclusion and also can to a large extent explain alternative views (invoking either indirect effects on other GTPases, or indirect effects from depletion of other GTPases to explain results suggesting that either the GAP activity of this family of RhoGAPs is not necessary or that it acts on other GTPases, in particular Rac as reported by [7]). The authors provide an elegant and impressive analysis of this issue, using a powerful combination of mutational analysis, live cell imaging and biochemistry. This manuscript provides the most extensive and compelling analysis of how CYK-4 influences cytokinesis via the RhoGEF ECT-2 and the RhoA GTPase, and should go a long ways toward resolving this complex issue.

While the manuscript in my opinion clearly warrants publication in *eLife*, the authors first need to address the following concerns that, while perhaps relatively minor, nevertheless could possibly improve the clarity of their arguments.

1) In the Introduction, subsection “The role of the GAP activity of CYK-4 GAP has been controversial”, the authors point out that CYK-4 has GAP activity toward RhoA but much more toward Rac1 and CDC-42. The authors should directly address how these observations relate to their conclusions concerning their model for how CYK-4 functions as an activator of RhoA.

2) With respect to Figure 1, the authors should discuss why there is some furrow ingression in embryos expressing CYK-4 lacking the C1 domain, and lacking *nop-1* and *ced-10* (in contrast to CYK-4^E448K^).

3) In two places (subsection “The *cyk-4(or749ts)* allele, E448K, exhibits defects in membrane association” and “CYK-4 GAP domain mutations that prevent RhoA binding are highly defective in RhoA activation”), the authors conclude results paragraphs by stating that an observation is “important” without explaining why it is important. Doing so might help clarify their conclusions.

4) In the subsection “Mutations in the active site of CYK-4 can be suppressed by RGA-3/4 depletion”, the authors refer to hyper-contractility after RGA-3/4 depletion and imply (and to some extent document) that this phenotype is suppressed by CYK-4 mutants. The authors should more explicitly state if this is an accurate conclusion.

5) In the subsection “Mutations in the active site of CYK-4 can be suppressed by RGA-3/4 depletion”, the authors refer to an allelic series of CYK-4 mutants. It might be helpful to explicitly state why the different mutants exhibit this gradation in effect.

6) In the first paragraph of the Discussion, the authors state that their data indicate that CYK-4 has to act catalytically on RhoA•GTP. The authors need to explain how their data support this conclusion.

7) Is it possible that the suppressor mutations in ECT-2 (*xs110* and *xs111*) affect other functions than GEF activity? It might be worth discussing this possibility.

8) In the subsection “Mechanism of ECT-2 activation and Positive Feedback”, the authors refer first to Figure 3, and then to Figures 2, 4 and 5 as supporting two conclusions. It would help if the authors elaborate to explicitly state what data in these figures support these conclusions.

9) The authors should explicitly state what they think the GAP active site does to influence ECT-2 GEF activity.

10) In the subsection “Mechanism of ECT-2 activation and Positive Feedback”, the authors state that CYK-4 GAP binds RhoA•GTP, but the model in Figure 7 in fact shows it binding RhoA:GDP. This should be explained.

11) The model diagram (Figure 7) did not really do much to help me understand how they authors think CYK-4 activates RhoA.

Reviewer #3:

CYK-4 plays a crucial role in cytokinesis as a non-motor subunit of centralspindlin, which bundles microtubules to form the central spindle and midbody. However, the role of the RhoGAP domain of CYK-4 remains unclear or controversial as the authors nicely describe in the Introduction. The cell biological and genetical data that the Glotzer lab has been providing in a recent series of papers are indeed very difficult to explain with Rac or Cdc42 as sole targets of CYK-4 but are consistent with the positive role of CYK-4 on RhoA activation. A major difficulty of this theory is that a GAP usually promotes conversion of a GTPase from GTP-form to GDP-form, which is in general an inactive form, by stimulating the GTP hydrolysis. A clear molecular mechanism for CYK-4 to activate RhoA has been a huge missing link in the file of cytokinesis.

In this manuscript, the authors examined various mutations in the GAP domain of CYK-4 including the E448K mutation that had previously been identified as a temperature-sensitive lethal allele *(or749ts)* and can be suppressed by loss-of-function of the CED-10/Rac or its effectors. By combination with the loss-of-function of NOP-1, a mysterious activator of Ect2 RhoGEF specific for nematodes, and RGA-3/4, a well conserved GAP that specifically inactivates Rho, the authors have strengthen their previous conclusion “that the RhoGAP domain of CYK-4 has an essential role in RhoA activation” (this is the title of [27]) except for a concern about the presence of other Rac proteins in addition to CED-10 (see below), which was not considered. Very interestingly, a screen for suppressors of *cyk-4(or749ts)* identified two novel alleles of Ect2 RhoGEF that cause its hyper activation, consistently with the proposed role of CYK-4 in RhoA activation. Finally, the authors tried to reveal biochemical mechanisms of RhoA activation by CYK-4 and found the direct binding between the GEF domain of ECT-2 and the C-terminal fragment of CYK-4 containing C1 and RhoGAP domains.

These are all interesting findings. However, it is questionable whether these are sufficient to support the authors’ rather strong claim that the non-canonical activity of CYK-4 RhoGAP domain promotes RhoA activation though its interaction with the GEF domain of ECT-2. The novel interaction between CYK-4 GAP and ECT-2 GEF domains relies solely on a single binding experiment in Figure 7 (7C and 7D are just unconvincing attempts to explain the hyper activation of ECT-2 by the new *ect-2* alleles suggested by the in vivo phenotype). I really appreciate the amount of work necessary for the cell biology and genetics experiments. However, these are just strengthening their own already published claim. To propose a molecular mechanism of RhoA activation by the CYK-4 RhoGAP domain, stronger in vitro data should be provided at lease those obvious and feasible ones such as the influences of all the GAP mutations studied in vivo on the CYK-4-ECT-2 interaction.

Another concern is that the authors' logics are sometimes inconsistent. For example, as the explanations for the similar weaker cortical accumulation of the E448K and EE mutants, which show very similar in vivo phenotypes (slow ingression and regression at ∼50% ingression around 300s, weaker cortical accumulation, no furrowing upon *nop-1* loss-of-function), they employed different reasonings, indirect effect on the function of C1 domain and the weakened interaction with RhoA, respectively. I agree with the authors on that mutations inside a protein fold might make the folding less stable more frequently than surface mutations and understand that a concern about this possibility prompted the current studies. However, this is an issue to be answered by proper biochemical/biophysical assays such as circular dichroism spectroscopy, which was undertaken in a recent paper from the Glotzer lab (White et al., 2013 http://www.ncbi.nlm.nih.gov/pmc/articles/PMC3707682/). Otherwise, based on the phenotypic similarity, it would be more reasonable to explain the phenotype of the E448K mutation is also caused by the reduced affinity to Rho (or other GTPases). Anyway, there are different possible mechanisms for the cortical recruitment of CYK-4 that are not necessarily exclusive to each other; C1-phosphoinositide interaction, GAP-Rho interaction, interaction with astral microtubule gathered by the ingressing furrow. This should be explained more explicitly rather than picking up a convenient one for individual cases.

Specific points:

1) A table of the *C. elegans* strains used in this study with strain names and genotypes should be provided.

2) The following statements are ambiguous: “The RhoGAP activity of CYK-4/MgcRacGAP functions non-canonically” (title), “serve as a RhoGAP” and “consistent with CYK-4 RhoGAP activity” (Abstract). It should be clarified what the authors mean by “RhoGAP”. ‘RhoGAP activity’ could be any kind of activity that the RhoGAP domain has, the GAP activity against any Rho-family GTPase or a GAP activity specifically for Rho (but not for other GTPases such as Rac or Cdc42). It might be helpful to use “RhoGAP” for a domain name and “Rho-GAP”, “Rac-GAP”, “Rho-family-GAP” etc. for referring.

3) In the subsection “RhoA activation during cytokinesis”, the authors state: “The direct activator of RhoA during cytokinesis is the RhoGEF ECT-2”. There are other RhoA activators during cytokinesis such as GEF-H1 (Birkenfeld J 2007 Dev. Cell http://www.ncbi.nlm.nih.gov/pubmed/17488622). “A major activator of RhoA…” would be reasonable.

4) In the subsection “RhoA activation during cytokinesis”, you claim that “recruitment of ECT-2 to the spindle midzone involves regulated binding between ECT-2 and CYK-4…”. So far, the regulation of the interaction between CYK-4 and ECT-2 by phosphorylation by PLK-1 has only been shown in mammalian cells although there is no strong reason to imagine that the same regulation might not work in *C. elegans*.

5) Please clarify the following reference: “…it appears to be required to activate RhoA (D'Avino 2004)”. Although D'Avino et al. claimed “these observations are consistent with RacGAP50C inhibiting Rac and promoting Rho activity” and put RacGAP50C upstream to Rho in their model, this does not necessarily mean RacGAP50C is “required” for RhoA activation.

6) “These results suggest that this mutation in CYK-4 affects more than the RhoGAP activity of CYK-4 or that CED-10/Rac1 is not the relevant target of the GAP domain, or both” (Introduction). This is neglecting another obvious possibility that there might be another Rac that works redundantly with CED-10. *C. elegans* has two additional Rac-related genes, RAC-2 and MIG-2.

7) “…Subsequent analysis demonstrated that *cyk-4(or749ts)* does cause a phenotype in *ced-10*-embryos (27)” (Introduction). This might be due to elevated RAC-2 and MIG-2 activities by the *cyk-4* mutation, which still remained after the depletion of CED-10. Imperfect restoration of the wild-type behavior does not discredit the importance of the suppression of Rac and downstream effectors.

8) Figures 1 and 3: Why is there a big difference between the central spindle signal of CYK-4^∆C1^::GFP and CYK-4^E448K^::GFP? Five slices with 2.5 µm intervals should cover a range of 10 µm, which covers more than two thirds of the thickness of the 50%-ingressed cleavage furrow. For quantification, it is not clear how the usage of mCherry::PH as a standard is justified. The PH domain labels the plasma membrane by binding to phosphoinositides, which might be regulated by Rho-fmaily GTPases (http://www.ncbi.nlm.nih.gov/pubmed/24914539, http://www.ncbi.nlm.nih.gov/pmc/articles/PMC4114633/). Indeed, mCherry::PH signal at the 50%-ingresssed furrow is stronger in *cyk-4*^*R459A*^*::gap* embryo (Figure 4) than in *cyk-4*^*WT*^ embryos (Figure 1 the first row, Figure 4). The intensities of mCherry::PH used for standardization must be presented.

9) Another issue related to these experiments is that how CYK-4 accumulates at the furrow cortex is not clear. Although the authors mention it as accumulation on the plasma membrane, it might be reflecting the interactions between the C1 domain and phosphoinositides or between RhoGAP-Rho, or the compaction of the equatorial asters by the ingressing furrow.

10) How about the temperature sensitivity of EE and R459A?

11) In the subsection “The *cyk-4(or749ts)* allele, E448K, exhibits defects in membrane association”, the authors state: “the onset of furrow ingression is delayed relative to controls in *cyk-4*^*E448K*^ embryos, but not in *cyk-4*^*∆C1*^ embryos”. In Figure 1, the deformation of the plasma membrane in the E448K embryo starts at the similar timing to the WT one (∼ at the 9th time point) while it becomes debatable in the ∆C1 embryo in the 11th or 12th time point. From the graph in Figure 1, no clear difference is detected in the timing of the furrow initiation while the ingression is clearly slower in the E448K embryos.

12) Please clarify the following passage: “…such that it facilitates the abscission step in *cyk-4(or749ts)*; *ced-10(n1933)* embryos”. It is not clear why the abscission was suddenly mentioned while no experiment has been performed to specifically examine it.

13) Figure 2—figure supplement 1: Controls are missing to prove that the bands in the bead-bound fractions represent the specific binding to RhoA instead of beads or GST. Binding to the GST-alone beads needs to be performed in parallel. How is the binding affected by the E448K mutation?

14) In the subsection “Mutants in the active site of the CYK-4 GAP domain are cytokinesis-defective”, the authors claim: “Thus, the GAP activity of CYK-4 is not essential for post embryonic development but…”. This is superficially true. However, there is another possibility that the mutant message/product might delay the consumption of the wild-type message/product derived from the heterozygous mother and kept the level of functional CYK-4 enough for post embryonic development (but not for maturation of germ line). The sentences “Thus, while early embryos require the GAP activity of CYK-4, this requirement is relaxed post embryonically” and “…as some cell types may not require the GAP activity of CYK-4, as seen in post embryonic cells in *C. elegans*” (subsection “GEF activation model accounts for previous results”) should also be reconsidered.

15) “CYK-4^R459A^ hyper accumulates on the membrane as compared to WT CYK-4; this localization suggests that CYK-4^R459A^ is well folded in vivo” – is this logical? In general, partial misfolding might result in tighter membrane binding by forming small aggregates.

16) The sentence: “If furrow formation is dependent on CYK-4 binding to either CED-10/Rac1 or CDC-42, then inactivation of these GTPases would be predicated to cause a phenotype at least as severe as a mutation that weakens the GTPase binding site of CYK-4”. This inexplicitly assumes that these GTPases are positive regulators of the furrow formation. Thus, the logical conclusion of experimental observations is that furrow formation is independent of CYK-4 binding to them or that these are not positive regulators of the furrow formation.

17) Figure 1–figure supplement 4 and 5: Is the structure on which CYK-4^E448K^::GFP was detected really the gonad membrane? The DIC image is not clear which structure is stained. Only after comparison with the images in Figure1–figure supplement 5 can readers understand what the structure labeled with the CYK-4 mutant in Figure 1—figure supplement 4 is.

18) Please clarify this statement: “These data demonstrate that the catalytic activity of the CYK-4 GAP domain must have a function that is distinct from maintaining CED-10/Rac1 in an inactive sate”. This assumes that CED-10 is the only GTPase to be inactivated by CYK-4.

19) “However, CYK-4 and ECT-2 form a protein complex through their regulatory N-termini, therefore the C-terminal RhoGAP domain of CYK-4 will be in the vicinity of the ECT-2 RhoGEF domain” (subsection “*cyk-4* suppressor mutations activate *ect-2*). This is not true. Whether the C-termini are in the vicinity is nothing to do with the fact that they bind through the N-termini.

20) In the same subsection: “…the E129K substitution, but not G707D, significantly reduces binding of the ECT-2 N- and C-terminal (Figure 7)…”. As the MBP-ECT-2_N wild type and E129K proteins are the different protein samples prepared independently, strictly, the data provided can't exclude a possibility that MBP-ECT-2_N wild-type shows stronger non-specific binding to the GSH-beads or GST than the E129K mutant. Control experiments with GST instead of GST-ECT-2_DHPH should be performed in parallel.

21) In the Results: “The ECT-2G707D variant exhibits a modest increase in GEF activity compared to wild-type ECT-2 over a range of concentrations (Figure 7).” This is not really convincing. The same amount of the wild-type and mutant proteins are supposed to have been loaded on the gel. However, there is a small but clear difference in the intensity of the bands, especially the ∼50 kDa minor ones. I am afraid that there might also be a similar difference between the main bands. The difference between 1 µg and 1.2 µg is barely detectable by SDS-PAGE and Coomassie staining. In general, it is not trivial to quantify protein concentration with the precision of less than 10% error. Anyway, in the first place, it is questionable whether such a minor difference in the GEF activity can explain the dramatic in vivo phenotype.

22) The references in this sentence are misleading: “Accurate reconstitutions will need to account for the facts that cytokinetic RhoA activation involves the CYK-4 C1 domain, the ability of CYK-4 to bind to ZEN-4, and the ability of ZEN-4 to oligomerize (Figure 1) (1).” Figure 1 does not test the oligomerization. Defect in the oligomerization does not prevent RhoA activation (Hutterer, 2010 http://www.ncbi.nlm.nih.gov/pubmed/19962307, Dr. Glotzer is one of the authors of this paper). [1] demonstrated elevated plasma membrane association of the centralspindlin mutant defective for 14-3-3-binding and thus promoted for oligomerization (*zen-4(S682A)*). However, it is not clear whether the rather global cortical contractility caused by this mutation is due to activation/inactivation of RhoA or other GTPases.

23) “RhoA inactivation (28)” (subsection “GEF activation model accounts for previous results”) – this is misleading, as the major claim of this paper is the importance of the flux of RhoA turnover facilitated by combination of GEF and GAP for the formation of a sharp zone of the active form of RhoA.

24) Legend for Figure 6—figure supplement 1: What was the temperature?

25) In the subsection “Image quantification”, it is not clear how the time-series data sets of the normalized cortical distance were analyzed by the Kruskal-Wallis test. Were the rates of ingression calculated and then tested? Or does the test directly compare the shape of the curves? Please provide details and proper references. Referring to the Kruskal-Wallis test by “ANOVA” (legends for Figures 2, 3, 4 and 1) is confusing since ANOVA is usually used to refer to (parametric) analysis of variance, which requires the normality of the data, and different from the Kruskal-Wallis test, which is based on ranks and does not make any assumption about the probability distributions of the variables.

26) Figure 1, Figure 1—figure supplement 2, Figure 2 and Figure 3: For the *t*-tests, which method of correction for multiple comparison was used?

---

## [Author Response]

Reviewer #1:

*[…] I think the paper is appropriate for publication in* eLife*. However, I would like to see the biochemistry developed better. In particular, I am surprised that none of the GAP mutants identified in the work have been characterized in the biochemical studies described in the latter part of the paper. The two key experiments are: 1) how do the GAP mutants behave in the physical interaction experiments with ECT2 and 2) how does the presence of wild-type or GAP mutants of CYK-4 affects the ECT2-mediated nucleotide exchange on RhoA. These experiments are crucial if they are to advance the model in*
Figure 7*.*

We have assayed the binding of the GAP mutants binding to ECT–2. These data are included in Figure 7—figure supplement 1 which demonstrates that the mutations do not affect the binding of CYK–4 GAP domain to the DH-PH domain of ECT–2.

I think some more discussion is also warranted on how mutations in the PH domain potentially affect the GEF activity. Molecular dynamics may help, although I realize this can be a new foray altogether.

The end of the Results section was amended by adding a concluding explanatory phrase: “The ECT–2G707D variant exhibits a modest increase in GEF activity compared to wild-type ECT–2 over a range of concentrations (Figure 7), perhaps by increasing the affinity of ECT–2 for RhoA.”

Reviewer #2:

*[…] While the manuscript in my opinion clearly warrants publication in* eLife*, the authors first need to address the following concerns that, while perhaps relatively minor, nevertheless could possibly improve the clarity of their arguments.*

1) In the Introduction, subsection “The role of the GAP activity of CYK-4 GAP has been controversial”, the authors point out that CYK-4 has GAP activity toward RhoA but much more toward Rac1 and CDC-42. The authors should directly address how these observations relate to their conclusions concerning their model for how CYK-4 functions as an activator of RhoA.

We have more extensively addressed the differential activity of CYK–4 GAP activity towards Rho family GTPases in the Discussion. We suggest that it may be inactive towards RhoA because it would induce futile cycling of RhoA. We now mention that the activity toward Rac activity could have other functions in other contexts: “Not only does this model explain why CYK–4 retains GAP activity towards RhoA, it also explains why its ability to inactivate RhoA is attenuated. High turnover rates of RhoA•GTP induced by CYK–4 might rapidly consume RhoA•GTP at the site of production, yielding a futile cycle of RhoA activation and inactivation.”

We have also added the statement (subsection “GEF activation model can account for many previous results”): “We do not rule out the possibility that, in certain contexts, CYK–4 or its orthologs negatively regulate Rac or Cdc42. Indeed, recent evidence indicates that the Xenopus ortholog of CYK–4 regulates junctional GTPases.”

*2) With respect to*
Figure 1*, the authors should discuss why there is some furrow ingression in embryos expressing CYK-4 lacking the C1 domain, and lacking* nop-1 *and* ced-10 *(in contrast to CYK-4*^*E448K*^*).*

In embryos expressing CYK–4^∆C1^ (Figure 1) we observe some furrowing which is due to the presence of NOP–1 as it is absent in the double mutant (Figure 1). The furrowing in *CYK–4*^*∆C1*^ embryos is slightly enhanced when CED–10 is mutated, however these embryos are not defective for NOP–1. We did not present data on CYK–4^∆C1^, *nop–1(-)* or *ced–10(-)*.

*3) In two places (subsection “The* cyk-4(or749ts) *allele, E448K, exhibits defects in membrane association” and “CYK-4 GAP domain mutations that prevent RhoA binding are highly defective in RhoA activation”), the authors conclude results paragraphs by stating that an observation is “important” without explaining why it is important. Doing so might help clarify their conclusions.*

We thank the reviewer for helping to clarify the manuscript. In the first case is it important because it shows that suppression of CYK–4 ^E448K^ by inactivation of *ced–10(-)* is completely dependent on NOP–1. That is, *nop–1(-)* and CYK–4^WT^ embryos are cytokinesis competent (and viable), but *nop–1(-)*, CYK–4^E448K^ and *ced–10(-)* are completely defective in furrowing which would not be the prediction if the sole function of the GAP activity of CYK–4 is to maintain CED–10 in the inactive state.

The revised text now reads: “Importantly, when NOP–1 activity is compromised, inactivation of CED–10/Rac1 does not suppress the defect in furrow ingression caused by CYK–4^E448K^ (Figure 1), suggesting that CYK–4 does not act directly on CED–10.”

Similarly in the second case, the result shows that whereas CYK–4^WT^ is capable of inducing furrow ingression in the absence of NOP–1 CYK–4EE is not. The revised text now reads: “Thus, Rho GTPase binding by CYK–4 is essential for centralspindlin-mediated cytokinetic ingression.”

4) In the subsection “Mutations in the active site of CYK-4 can be suppressed by RGA-3/4 depletion”, the authors refer to hyper-contractility after RGA-3/4 depletion and imply (and to some extent document) that this phenotype is suppressed by CYK-4 mutants. The authors should more explicitly state if this is an accurate conclusion.

We did not intend to imply that all of the hypercontractility observed in RGA–3/4-depleted embryos is CYK–4 dependent. Indeed, in a previous publication we showed that the hypercontractility *during pseudocleavage* in RGA–3/4-depleted embryos is NOP–1 dependent. Thus RGA–3/4 depletion enhances RhoA activity, irrespective of the activator – which in pseudocleavage is NOP–1 and in cytokinesis is primarily CYK–4. We modified this sentence to clarify this point without getting into too much detail: “As previously shown, depletion of RGA–3/4 causes cortical hypercontractility in otherwise wild-type embryos, *during both pseudocleavage and during cytokinesis*, and results in embryonic lethality (Figure 5) (32; 33)”. In addition, we added this to the Discussion: “However, loss of CYK–4 GAP activity and loss of RGA–3/4 counterbalance each other *during cytokinesis*”.

5) In the subsection “Mutations in the active site of CYK-4 can be suppressed by RGA-3/4 depletion”, the authors refer to an allelic series of CYK-4 mutants. It might be helpful to explicitly state why the different mutants exhibit this gradation in effect.

6) In the first paragraph of the Discussion, the authors state that their data indicate that CYK-4 has to act catalytically on RhoA•GTP. The authors need to explain how their data support this conclusion.

These are two excellent suggestions, which we have addressed by adding a supplemental figure (Figure 7—figure supplement 2) and by adding a paragraph to the Discussion (“This working model is supported by the finding that full activation […] within the ECT– 2 GEF domain”).

*7) Is it possible that the suppressor mutations in ECT-2 (*xs110 *and* xs111*) affect other functions than GEF activity? It might be worth discussing this possibility.*

Two lines of evidence indicate that these mutations increase the overall activity of ECT–2. First, the mutations cause an increase in contractility. Second, biochemical evidence indicates that *xs111* impairs the auto inhibitory binding between the N-and C-termini of ECT–2 and that *xs110* causes a modest increase in GEF activity. While circumstantial this interpretation is consistent with the partial suppression caused by depletion of RGA–3/4.

*8) In the subsection “Mechanism of ECT-2 activation and Positive Feedback”, the authors refer first to*
Figure 3*, and then to*
Figures 2, 4 and 5
*as supporting two conclusions. It would help if the authors elaborate to explicitly state what data in these figures support these conclusions.*

Please see points 5 and 6 above.

9) The authors should explicitly state what they think the GAP active site does to influence ECT-2 GEF activity.

Please see points 5 and 6 above.

*10) In the subsection “Mechanism of ECT-2 activation and Positive Feedback”, the authors state that CYK-4 GAP binds RhoA•GTP, but the model in*
Figure 7
*in fact shows it binding RhoA:GDP. This should be explained.*

Please see points 5 and 6 above.

*11) The model diagram (*Figure 7*) did not really do much to help me understand how they authors think CYK-4 activates RhoA.*

We hope the additional paragraph and supplemental figure make our model more explicit and clarify this point (please see points 5 and 6, above).

Reviewer #3:

Specific points:

*1) A table of the* C. elegans *strains used in this study with strain names and genotypes should be provided.*

Our strain table has been provided.

2) The following statements are ambiguous: “The RhoGAP activity of CYK-4/MgcRacGAP functions non-canonically” (title), “serve as a RhoGAP” and “consistent with CYK-4 RhoGAP activity” (Abstract). It should be clarified what the authors mean by “RhoGAP”. ‘RhoGAP activity’ could be any kind of activity that the RhoGAP domain has, the GAP activity against any Rho-family GTPase or a GAP activity specifically for Rho (but not for other GTPases such as Rac or Cdc42). It might be helpful to use “RhoGAP” for a domain name and “Rho-GAP”, “Rac-GAP”, “Rho-family-GAP” etc. for referring.

We have clarified the substrate for the RhoGAP activity in the Abstract. In addition, where possible, we have used the agnostic term GAP domain in many places and where we discuss RGA–3/4, we refer to it as a RhoA GAP.

*3) In the subsection “RhoA activation during cytokinesis”, the authors state: “The direct activator of RhoA during cytokinesis is the RhoGEF ECT-2”. There are other RhoA activators during cytokinesis such as GEF-H1 (Birkenfeld J 2007 Dev. Cell*
*http://www.ncbi.nlm.nih.gov/pubmed/17488622**). “A major activator of RhoA…” would be reasonable.*

ECT–2 is required for all RhoGAP activity during cytokinesis in human cells and *C. elegans*. We have revised this sentence to read: “The primary direct activator of RhoA during cytokinesis is the RhoGEF ECT–2”.

*4) In the subsection “RhoA activation during cytokinesis”, you claim that “recruitment of ECT-2 to the spindle midzone involves regulated binding between ECT-2 and CYK-4…”. So far, the regulation of the interaction between CYK-4 and ECT-2 by phosphorylation by PLK-1 has only been shown in mammalian cells although there is no strong reason to imagine that the same regulation might not work in* C. elegans*.*

The reviewer is correct about this point. We do not wish to imply that this has been established for the *C. elegans* proteins, though, like the reviewer, we have no reason to believe that it is not the case for the following reasons. First, preliminary experiments suggested that Plk1 phosphorylation of *C. elegans* CYK–4 facilitates binding by *C. elegans* ECT–2. Secondly, Plk–1 depletion has been shown to impair completion of cytokinesis (PMID 20823068). Finally, in the sentences that precede this remark, we state: “Though these proteins are conserved, their names are distinct in each organism. For simplicity, *C. elegans* names will be used throughout this manuscript with the exception that we will refer to RHO–1 with the more common name RhoA.” Thus, it is explicit that the statement in question does not necessarily refer to the nematode proteins and, indeed, the cited references refer to experiments in cultured mammalian cells.

*5) Please clarify the following reference: “…it appears to be required to activate RhoA (*[10]*)”. Although D'Avino et al. claimed “these observations are consistent with RacGAP50C inhibiting Rac and promoting Rho activity” and put RacGAP50C upstream to Rho in their model, this does not necessarily mean RacGAP50C is “required” for RhoA activation.*

We have edited this phrase to “it appears to promote RhoA activation”. In addition, the previous sentence also cites [10] and states: “in others it appears to be important to negatively regulate Rac1”.

*6) “These results suggest that this mutation in CYK-4 affects more than the RhoGAP activity of CYK-4 or that CED-10/Rac1 is not the relevant target of the GAP domain, or both” (Introduction). This is neglecting another obvious possibility that there might be another Rac that works redundantly with CED-10.* C. elegans *has two additional Rac-related genes, RAC-2 and MIG-2.*

We thank the reviewer for raising this point. We have addressed this point by a number of avenues. First, in the literature, [7], reported:

“RNAi of CDC–42 or MIG–2 also did not ameliorate the CYK–4GAP(E448K) cytokinesis phenotype (Figure 3). However, RNAi of CED–10 or RAC–2 led to substantial rescue, allowing 70% and 24%, respectively, of CYK–4GAP(E448K) embryos to successfully complete the first cytokinesis. Simultaneous RNAi of CED–10 and RAC–2 did not increase the efficiency of rescue over RNAi of RacCED–10 alone (Figure Supplement 4D).”

Thus previous work does not support a cytokinetic role for RAC–2 or MIG–2. However, as the authors noted in the supplemental information, the conservation between CED–10 and RAC–2 at the nucleotide level are such that either of these dsRNAs could, in principle, deplete both CED–10 and RAC–2.

To clarify this issue, several years ago, we examined whether a strong loss of function mutation in *ced–10* could suppress the cytokinetic defect in *cyk–4(or749ts)* and found that it could (27). This strongly suggested that CED–10 is the major Rac protein whose activity modulates cytokinesis. If RAC–2 had a significant role, mutation of *ced–10* would not be predicted to phenocopy the potential co-depletion of both CED–10 and RAC–2. Furthermore, mutation of *ced–10* and depletion of the *arp2/3* subunit ARX–2 identically modified the phenotype *cyk–4(or749ts)* (27). If both CED–10 and RAC–2 functioned upstream of *arx–2*, mutation of *ced–10* would be predicted to be less effective than depletion of ARX–2.

To further examine whether other Rac proteins will modulate cytokinesis in the early embryo, we depleted either ARX–2 or RAC–2 in *cyk–4*^*R459A*^ embryos. We found that depletion of ARX–2, but not RAC–2, rescued cytokinesis defect (Figure 4—figure supplement 1). This suggests that RAC–2 is not a target for the GAP domain of CYK–4.

Furthermore, *mig-2* gain of function alleles (mutations in the equivalent of G12 of Ras) are viable, though they exhibit cell migration defects (52). These gain of function phenotypes are significantly stronger than the corresponding null mutation. This indicates that the *mig–2* alleles are indeed activated alleles and the high viability of these strains suggests that *mig–2* does not regulate cytokinesis. To further investigate the possibility that *mig–2* does affect cytokinesis, we obtained a gain of function allele *mig–2(gm103)* and tested whether this allele affects cytokinesis in NOP–1 or ZYG–9 depleted embryos (Figure 4—figure supplement 3). These two depletions were chosen because they are effective ways of sensitizing cytokinesis to genetic perturbations (see PMID 17669650, 22918944 for examples). Unlike *cyk–4*^*R459A*^ embryos, *mig– 2*;*nop–1* embryos successfully finish cytokinesis, indicating that activation of MIG–2 does not affect the centralspindlin-dependent pathway for furrow formation. In addition, *mig–2*;*zyg–9* embryos form both anterior and posterior furrows, indicating that the central spindle-independent and central spindle-dependent pathways are not affected by activation of MIG–2.

Next, we depleted either ARX–2 or RAC–2 from *nop–1;ced–10 cyk–4(or749ts)* embryos. We found that neither arx–2 nor rac–2 depletion modified the cytokinesis phenotypes in *nop–1;ced–10;cyk–4(or749ts)* embryos, i.e. *arx–2 nop–1;ced–10 cyk– 4(or749ts)* and *rac–2;nop–1;ced–10 cyk–4(or749ts)* embryos had no furrow ingression during cytokinesis (Figure 4—figure supplement 2). These results suggest that RAC–2, nor any other proteins upstream of ARX–2, are critical targets for the GAP activity of CYK–4.

We note that previous studies have also concluded that: “by contrast, *ced–10* is uniquely required for cell-corpse phagocytosis, and *mig–2* and *rac–2* have only subtle roles in this process. (PMID 11714673)”.

Thus, there are precedents for different Rac proteins in *C. elegans* functioning both redundantly and non-redundantly.

*7) “…Subsequent analysis demonstrated that* cyk-4(or749ts) *does cause a phenotype in* ced-10*-embryos (*[27]*)” (Introduction). This might be due to elevated RAC-2 and MIG-2 activities by the* cyk-4 *mutation, which still remained after the depletion of CED-10. Imperfect restoration of the wild-type behaviour does not discredit the importance of the suppression of Rac and downstream effectors.*

Please see point 6.

*8)*
Figures 1 and 3*: Why is there a big difference between the central spindle signal of CYK-4*^*∆C1*^*::GFP and CYK-4*^*E448K*^*::GFP? Five slices with 2.5 µm intervals should cover a range of 10 µm, which covers more than two thirds of the thickness of the 50%-ingressed cleavage furrow. For quantification, it is not clear how the usage of mCherry::PH as a standard is justified. The PH domain labels the plasma membrane by binding to phosphoinositides, which might be regulated by Rho-fmaily GTPases (**http://www.ncbi.nlm.nih.gov/pubmed/24914539**,*
*http://www.ncbi.nlm.nih.gov/pmc/articles/PMC4114633/**). Indeed, mCherry::PH signal at the 50%-ingresssed furrow is stronger in* cyk-4^R459A^::gap *embryo (*Figure 4
*ii) than* in cyk-4^WT^
*embryos (*Figure 1
*the first row,*
Figure 4*). The intensities of mCherry::PH used for standardization must be presented.*

*E448K* is a Ts allele that retains central spindle assembly function, but it is not surprising that this temperature-sensitive mutation may not accumulate with wild-type efficiency at the restrictive temperature.

9) Another issue related to these experiments is that how CYK-4 accumulates at the furrow cortex is not clear. Although the authors mention it as accumulation on the plasma membrane, it might be reflecting the interactions between the C1 domain and phosphoinositides or between RhoGAP-Rho, or the compaction of the equatorial asters by the ingressing furrow.

We agree with the reviewer’s assessment that the exact mechanism by which CYK–4 accumulates at the furrow cortex is not entirely clear. However, the results in this manuscript suggest that membrane association of CYK–4 involves both its C1 domain and its ability to bind RhoA. In addition, our results indicate that it is not mediated by the compaction of the astral microtubules because its membrane localization domain is required, which would not be predicted to (nor are they seen to) affect central spindle assembly.

10) How about the temperature sensitivity of EE and R459A?

CYK–4^EE^ and CYK–4^R459A^ fail to support viability at both 16°C and 25°C (Figure 3—figure supplement 1).

*11) In the subsection “The* cyk-4(or749ts) *allele, E448K, exhibits defects in membrane association”, the authors state: “the onset of furrow ingression is delayed relative to controls in* cyk-4^E448K^
*embryos, but not in* cyk-4^∆C1^
*embryos”. In*
Figure 1*, the deformation of the plasma membrane in the E448K embryo starts at the similar timing to the WT one (∼ at the 9th time point) while it becomes debatable in the ∆C1 embryo in the 11th or 12th time point. From the graph in*
Figure 1*, no clear difference is detected in the timing of the furrow initiation while the ingression is clearly slower in the E448K embryos.*

The initial furrowing the reviewer is referring to in the case of *E448K* is so slight that it does not show up in the graphs. We have added the work “significant” prior to furrow ingression.

*12) Please clarify the following passage: “…such that it facilitates the abscission step in* cyk-4(or749ts); ced-10(n1933) *embryos”. It is not clear why the abscission was suddenly mentioned while no experiment has been performed to specifically examine it.*

Abscission is mentioned because of the evidence from the Petronczki lab on the function of the C1 domain in human cells. If the defect in ∆C1 was very similar to that of *cyk–4(or749) [E448K]* then both mutants should be suppressed by ced–10 mutation, but suppression is only seen with *cyk–4(or749).* The sentence in question has been modified.

*13)*
Figure 2—figure supplement 1*: Controls are missing to prove that the bands in the bead-bound fractions represent the specific binding to RhoA instead of beads or GST. Binding to the GST-alone beads needs to be performed in parallel. How is the binding affected by the E448K mutation?*

We have repeated these experiments to provide the requested controls (Figure 2—figure supplement 1). However, as these additional controls render the figure significantly more complicated, we have not replaced the supplemental figure (though it would be straightforward to do if the editor/reviewers deemed it appropriate). The EE mutant serves as an internal control for CYK–4 GAP domains non-specifically binding to GST. We have not investigated the binding to RhoA by *E448K* allele because (i) this allele is not a primary focus of the manuscript, it only features in the manuscript to show that it is not an appropriate allele for specifically assessing GAP function and as an tool to obtain suppressor alleles; (ii) as a Ts allele, its behavior is expected to be temperature dependent and it is not uncommon for Ts alleles to exhibit more severe defect in vitro.

*14) In the subsection “Mutants in the active site of the CYK-4 GAP domain are cytokinesis-defective”, the authors claim: “Thus, the GAP activity of CYK-4 is not essential for post embryonic development but…”. This is superficially true. However, there is another possibility that the mutant message/product might delay the consumption of the wild-type message/product derived from the heterozygous mother and kept the level of functional CYK-4 enough for post embryonic development (but not for maturation of germ line). The sentences “…while early embryos require the GAP activity of CYK-4, this requirement is relaxed post embryonically” and “…as some cell types may not require the GAP activity of CYK-4, as seen in post embryonic cells in* C. elegans*” (subsection “GEF activation model accounts for previous results”) should also be reconsidered.*

While it is formally possible that zygotic rescue is due to the indirect mechanism the review alludes to, however, we think this is very unlikely as equivalent zygotic rescue is not observed with CYK–4^∆C1^ which would be predicted to rescue via similar indirect effects.

*15) “CYK-4*^*R459A*^
*hyper accumulates on the membrane as compared to WT CYK-4; this localization suggests that CYK-4*^*R459A*^
*is well folded in vivo” – is this logical? In general, partial misfolding might result in tighter membrane binding by forming small aggregates.*

As R459 is a charged residue on the surface of the protein, there is no reason a priori why it would cause misfolding. Furthermore, a misfolded protein would not be predicted to localize in a highly specific site in a cell cycle specific manner. Finally, this type of behavior has been observed previously with the RhoGAP RGA–3/4 (see Figure 1, [46] PMID 24012485).

16) The sentence: “If furrow formation is dependent on CYK-4 binding to either CED-10/Rac1 or CDC-42, then inactivation of these GTPases would be predicated to cause a phenotype at least as severe as a mutation that weakens the GTPase binding site of CYK-4”. This inexplicitly assumes that these GTPases are positive regulators of the furrow formation. Thus, the logical conclusion of experimental observations is that furrow formation is independent of CYK-4 binding to them or that these are not positive regulators of the furrow formation.

The reviewer is correct that this is an implicit assumption of the experiment. To address this concern, we have made this assumption explicit: “If furrow formation is dependent on CYK–4 binding to either CED–10/Rac1 or CDC–42 to generate a positive regulatory complex, then inactivation of these GTPases would be predicted to cause a phenotype at least as severe as a mutation that weakens the GTPase binding site of CYK–4.”

*17) Figure 1–figure supplement 4 and 5: Is the structure on which CYK-4*^*E448K*^*::GFP was detected really the gonad membrane? The DIC image is not clear which structure is stained. Only after comparison with the images in Figure1–figure supplement 5 can readers understand what the structure labeled with the CYK-4 mutant in Figure 1–figure supplement 4 is.*

To clarify this point, we have merged Figure 1–figure supplements 4 and 5.

18) Please clarify this statement: “These data demonstrate that the catalytic activity of the CYK-4 GAP domain must have a function that is distinct from maintaining CED-10/Rac1 in an inactive sate”. This assumes that CED-10 is the only GTPase to be inactivated by CYK-4.

See point 6 above.

*19) “However, CYK-4 and ECT-2 form a protein complex through their regulatory N-termini, therefore the C-terminal RhoGAP domain of CYK-4 will be in the vicinity of the ECT-2 RhoGEF domain” (subsection “*cyk-4 *suppressor mutations activate* ect-2*). This is not true. Whether the C-termini are in the vicinity is nothing to do with the fact that they bind through the N-termini.*

The issue here appears to be a semantic one related to the definition of vicinity. When two proteins form a complex, they are in the vicinity of each other (i.e. within ∼ 10 nm, depending on the size/shape of the proteins). Assuming the various domains are connected by flexible linkers, the interaction of one domain on each of the two proteins would greatly favor the interaction with the other, by virtue of increasing their effective concentration. This is described in detail in an excellent review by Kuriyan and Eisenberg (PMID 18075577).

*20) In the same subsection: “…the E129K substitution, but not G707D, significantly reduces binding of the ECT-2 N- and C-terminal (*Figure 7*)…”. As the MBP-ECT-2_N wild type and E129K proteins are the different protein samples prepared independently, strictly, the data provided can't exclude a possibility that MBP-ECT-2_N wild-type shows stronger non-specific binding to the GSH-beads or GST than the E129K mutant. Control experiments with GST instead of GST-ECT-2_DHPH should be performed in parallel.*

We have performed this additional control experiment which demonstrated that the interaction is not mediated via the GST domain (Figure 7—figure supplement 1). Again, we leave the original simpler figure as the E129K mutant provides and internal control that MBP-ECT–2N constructs do not non-specifically interact with GST tagged ECT–2C constructs.

*21) In the Results: “The ECT-2G707D variant exhibits a modest increase in GEF activity compared to wild-type ECT-2 over a range of concentrations (*Figure 7*)”. This is not really convincing. The same amount of the wild-type and mutant proteins are supposed to have been loaded on the gel. However, there is a small but clear difference in the intensity of the bands, especially the ∼50 kDa minor ones. I am afraid that there might also be a similar difference between the main bands. The difference between 1 µg and 1.2 µg is barely detectable by SDS-PAGE and Coomassie staining. In general, it is not trivial to quantify protein concentration with the precision of less than 10% error. Anyway, in the first place, it is questionable whether such a minor difference in the GEF activity can explain the dramatic in vivo phenotype.*

This mutation certainly provides rescue of both *cyk–4(or749)* and *cyk-4*^*R459A*^; structural homology modeling suggests this mutation lies near the Rho binding interface and a reproducible increase in specific activity is observed. The reviewer questions whether this change is specific activity is sufficient to explain the phenotype in vivo. Given the available information, the simplest interpretation is that the mutation increase GEF activity. It is possible that the in vivo assay in dilute aqueous solutions doesn’t fully capture the effect that this mutation has in vivo.

*22) The references in this sentence are misleading: “Accurate reconstitutions will need to account for the facts that cytokinetic RhoA activation involves the CYK-4 C1 domain, the ability of CYK-4 to bind to ZEN-4, and the ability of ZEN-4 to oligomerize (*Figure 1*) (*[1]*).”*
Figure 1
*does not test the oligomerization. Defect in the oligomerization does not prevent RhoA activation (Hutterer, 2010*
*http://www.ncbi.nlm.nih.gov/pubmed/19962307**, Dr. Glotzer is one of the authors of this paper).*
[1]
*demonstrated elevated plasma membrane association of the centralspindlin mutant defective for 14-3-3-binding and thus promoted for oligomerization (zen-4(S682A)). However, it is not clear whether the rather global cortical contractility caused by this mutation is due to activation/inactivation of RhoA or other GTPases.*

The citations were collected at the end of the sentence for clarity. They have been relocated within the sentence to address this reviewer’s concerns. Although the reviewer is correct that mutation in the oligomerization domain does not prevent furrowing, it is important to note that the defect in oligomerization motif was assayed in the presence of NOP–1, which we have shown is sufficient to induce furrowing in *cyk–4* mutant embryos. However, this point is addressed by Basant et al. In that paper we showed that *nop–1;air–2* embryos are defective in contractility and this defect can be completely reversed by depletion of par–5. Furthermore, the hypercontractility phenotype caused by depletion of PAR–5 is phenocopied by mutation of ZEN–4 S682A (which is unable to bind PAR–5). PAR–5 was previously shown to inhibit centralspindlin oligomerization by binding to phospho-S682 MKLP1 (PMID 20451386). Collectively, these results indicate that the defect in RhoA activation in *nop–1; air–2* mutants is caused by a failure of centralspindlin to oligomerize. Furthermore, in Basant et al., we showed that the hypercontractility caused by PAR–5 depletion (which is phenocopied by mutation of ZEN–4 S682A (Figure 6) is ECT–2-dependent (Figure 1) and is associated with an increase in RhoA activation (Figure 2). In summary, hyperoligomerization of ZEN–4 (either PAR–5 depletion or S682A) induces hyperactivation of RhoA and hypo-oligomerization of ZEN–4 (aurora B inhibition) prevents RhoA activation.

*23) “RhoA inactivation (*[28]*)” (subsection “GEF activation model accounts for previous results”) – this is misleading, as the major claim of this paper is the importance of the flux of RhoA turnover facilitated by combination of GEF and GAP for the formation of a sharp zone of the active form of RhoA.*

While the authors of this paper interpret their results as indicating the importance of flux, they did not directly measure flux. They did demonstrate that inactivation of the GAP activity of CYK–4 results in an increase in RhoA activation. Indeed the Abstract of that paper states: “We propose that the GAP domain of MgcRacGAP has two unexpected roles throughout cytokinesis: first, it transiently anchors active Rho, and second, it promotes local Rho inactivation, resulting in the constant flux of Rho through the GTPase cycle”.

24) Legend for Figure 6–figure supplement 1: What was the temperature?

These experiments were performed at 25°C.

*25) In the subsection “Image quantification”, it is not clear how the time-series data sets of the normalized cortical distance were analyzed by the Kruskal-Wallis test. Were the rates of ingression calculated and then tested? Or does the test directly compare the shape of the curves? Please provide details and proper references. Referring to the Kruskal-Wallis test by “ANOVA” (legends for*
Figures 2, 3, 4 and 1*) is confusing since ANOVA is usually used to refer to (parametric) analysis of variance, which requires the normality of the data, and different from the Kruskal-Wallis test, which is based on ranks and does not make any assumption about the probability distributions of the variables.*

We thank the reviewer for calling our attention to this issue. We have revised the manner in which we present the ingression rates. Rather than plotting the data with S.E.M., we instead provide 95% confidence intervals for each point. Thus, from inspection, the reader can directly observe that the data sets are distinct.

*26)*
Figure 1*,*
Figure 1—figure supplement 2*,*
Figure 2
*and*
Figure 3*: For the* t*-tests, which method of correction for multiple comparison was used?*

Figure 1, Figure 1—figure supplement 2 and Figure 3 were analyzed by one way ANOVA followed by Tukey multiple comparison. Figure 1, Figure 1—figure supplement 2, Figure 2, has only two data sets, so it was analyzed by *t*-test.